# The Fetal Region-specific Optimized Growth Standard (FROGS)—A fetal and birthweight centile calculator validated in a national population

Natasha L. Pritchard[1,2]*, Stephen Tong[1,2], Teresa MacDonald[1,2],
Elizabeth McCarthy[1,2], Lisa Hui[1,2,3,4], Michael Bethune[1,2], Hannah G. Gordon[1,2],
Roxanne Hastie[1,2], Emerson Keenan[1,2], Michael Permezel[1,2], Susan P. Walker[1,2‡],
Anthea C. Lindquist[1,2‡]

1 Department of Obstetrics, Gynaecology and Newborn Health, University of Melbourne, Melbourne,
Victoria, Australia, 2 Mercy Perinatal, Mercy Hospital for Women, Heidelberg, Victoria, Australia,
3 Reproductive Epidemiology Group, Murdoch Children's Research Institute, Parkville, Victoria, Australia,
4 The Northern Hospital, Epping, Victoria, Australia

‡ These authors share equal senior authorship on this work.
* natasha.pritchard@unimelb.edu.au, natashalpritchard@gmail.com

of Manchester, UNITED KINGDOM OF GREAT
BRITAIN AND NORTHERN IRELAND

**Peer Review History:** PLOS recognizes the
benefits of transparency in the peer review
process; therefore, we enable the publication
of all of the content of peer review and
author responses alongside final, published
articles. The editorial history of this article is
available here: https://doi.org/10.1371/journal.
pmed.1004634

## Abstract

### Background

There is no universally agreed upon obstetric growth standard for use during pregnancy. We aimed to design a simple novel growth standard, which incorporates key beneficial features identified in prior research.

### Methods and findings

We developed the Fetal Region-specific Optimized Growth Standard (FROGS), then validated it following International Federation of Gynaecology and Obstetrics (FIGO) guidelines. FROGS follows the shape of the fetal (ultrasound-based) Hadlock curve. It is *region-specific*; allowing adjustment for the mean birthweight and standard deviation of babies born at term in the local population where it will be applied. It provides an exact centile for each gestational day (rather than rounding off by weeks) and is optionally adjustable for fetal sex. Further, FROGS provides an 'estimate range' for the estimated fetal weight centile, assuming a 10% ultrasound measurement error. Following development, we validated FROGS in a retrospective cohort study by comparing its ability to identify small babies with an increased risk of adverse perinatal outcomes to four charts in current use: (1) population birthweight chart (Australian Institute of Health and Welfare, AIHW chart); (2) Hadlock's 1991 fetal chart; (3) Mikolajczyk's global fetal and birthweight centile chart; and (4) INTERGROWTH-21st fetal growth standards. To do this, we identified infants classified as small for gestational age (<10th centile) by each chart. We then identified non-overlapping <10th

**Data availability statement:** Data cannot be shared publically because of privacy restrictions for the women and babies included in our study. Data is available from the Consultative Council on Obstetric and Paediatric Mortality and Morbidity (CCOPMM) (contact consultative.councils@safercare.vic.gov.au). The coding and calculator used for FROGS can be accessed via https://github.com/centilechart/frogs.

**Funding:** NP is the recipient of a Norman Beischer Innovation Grant (https://nbmrf.org.au/grants/) and an Avant Doctors in Training Postgraduate Scholarship (https://avant.org.au/foundation/member-grants). TM is the recipient of an NHMRC EL1 Fellowship, which provides salary support, and a Norman Beischer Medical Research Foundation Clinical Research Fellowship (https://nbmrf.org.au/grants/). The funders had no role in study design, data collection and analysis, decision to publish, or preparation of the manuscript.

**Competing interests:** I have read the journal's policy and the authors of this manuscript have the following competing interests: NP is a recipient of a Norman Beischer Innovation Grant for this project. RH is a paid statistical consultant on PLOS Medicine's statistical board.

**Abbreviations:** AIHW, Australian Institute of Health and Welfare; FROGS, Fetal Region-specific Optimized Growth Standard; FIGO, International Federation of Gynaecology and Obstetrics; NICU, Neonatal Intensive Care Unit.

centile populations, i.e., infants classified as small by one chart, but not another. We compared rates of stillbirth and adverse perinatal outcomes between the non-overlapping populations. All charts except INTERGROWTH classified similar proportions of infants as <10th centile (10.4% FROGS, 9.3% AIHW, 11.1% Hadlock, 10.9% global, 4.4% INTERGROWTH). Of the three charts that classified similar proportions as <10th centile, infants classified by FROGS were at the highest risk of adverse perinatal outcomes. The infants classified as <10th centile by only FROGS had significantly increased relative risk (RR) of stillbirth, compared to the infants classified as <10th centile by only AIHW (RR 13.1, 95% CI 6.5–26.5), only Hadlock (RR 2.1, 95% CI 1.28–3.56) or only the global chart (RR 1.54, 95% CI 1.00–2.37). The FROGS chart outperformed these three charts in identifying infants at risk of other adverse perinatal outcomes associated with being small for gestational age, such as neonatal intensive care admission, Apgar scores <7 at 5 min, and operative (instrumental) vaginal birth for suspected fetal compromise. The cohort of infants classified as small for gestational age by INTERGROWTH was, in size and risk, closer to the cohort classified as <3rd centile by FROGS (3.4% of infants <3rd). This study is limited in that it retrospectively assesses birthweight, which may have different implications to a prospective evaluation of estimated fetal weight.

## Conclusions

Compared to currently used charts, the Fetal Region-specific Optimized Growth Standard outperforms existing charts that classify a similar proportion of infants as small for gestational age in identifying small infants at increased risk of stillbirth and other serious perinatal outcomes. The FROGS centile algorithm is simple and transparent. It has the potential to be adapted to other local populations, or applied to clinical and research settings globally.

---

### Author summary

#### Why was this study done?

- Growth centile curves are used in obstetrics, to identify small (particularly <10th centile) or large fetuses at risk of complications.

- However, the optimal growth standard is still heavily debated, with many varied growth centile curves in current use.

#### What did the researchers do and find?

- We designed and published the FROGS algorithm, which is an ultrasound based growth curve that can be adjusted for the mean birthweight and standard deviation of any given population, provides an exact centile for each gestational day,

and is optionally adjustable for fetal sex. Prior research has identified that incorporating these features best identifies small infants at risk of complications.

- We compared FROGS to four other commonly used growth standards (Hadlock, a global fetal and birthweight centile, Australian population charts, and INTERGROWTH international charts).

- We found that FROGS classified similar proportions of infants <10th centile as Hadlock, the global fetal and birthweight centile chart, and Australian population charts. When comparing non-overlapping populations (infants considered small by one chart but not another), FROGS outperformed these charts in identifying infants at risk of stillbirth and other adverse outcomes including small for gestational age, NICU admissions, low Apgar scores and operative birth (cesarean section or instrumental birth) for suspected fetal compromise.

- INTERGROWTH classified only 4.4% of infants below the 10th centile. The cohort of infants classified below the 10th centile by INTERGROWTH was, in size and risk, closer to the cohort classified below the 3rd centile by FROGS (3.4% of infants).

### What do these findings mean?

- Compared to currently used charts that classify a similar proportion of infants as <10th centile, FROGS may better identify small infants at increased risk of stillbirth or other serious complications.

- The FROGS algorithm can be adapted to other local populations or applied to global research settings.

- The findings of this study might differ, if FROGS was applied to estimated fetal weights measured on an ultrasound in pregnancy, rather than to known birthweights after delivery.

## Introduction

Growth centile curves are widely used in obstetrics, to identify small or large fetuses at risk of complications during pregnancy and birth, and to classify an infant's size after delivery [1,2]. Infants that are small for gestational age (<10th centile) are considered higher risk of stillbirth [3–6], obstetric and perinatal complications [7–12], and health issues into adulthood [10,13–17]. Consequently, infants born <10th centile are targeted for increased surveillance in pregnancy, and timed birth [18]. The growth curve used to classify infants as small or large therefore provides the foundation for all size-related management decisions [18,19].

Unfortunately, obstetric growth charts vary substantially between institutions throughout Australia, and worldwide [19]. If a fetus is incorrectly classified as small or large, there are significant downstream impacts on management. Variable classification is a barrier to standardized clinical care. Utilizing the most appropriate obstetric growth standard is therefore critically important.

Traditional birthweight charts are derived from all infants born at any given gestational age. However, infants born preterm are more likely to be growth restricted [20–22]. Therefore, their average birthweights are significantly lower than the average weight of healthy growing fetuses still in utero [23]. If we derive growth curves from infants born preterm, it risks underdiagnosing small for gestational age infants in a preterm population [24].

To overcome the issue inherent in birthweight charts, 'fetal' standards derive growth curves from ultrasonographic estimated weights of all healthy growing fetuses at any given gestation [25,26]. As in utero fetuses represent a healthier cohort compared to preterm-born infants, ultrasound-based growth charts are recommended by international guidelines [18,27]. One of the most widely used ultrasound-based charts is that developed by Hadlock and colleagues in Texas in 1991, derived from 392 middle-class, white women [25]. However, Hadlock may not adequately represent today's

multi-ethnic obstetric populations. International guidelines recommend that an obstetric growth curve represents the local population it is used in if traditional centile thresholds (such as <10th centile) are to be used [18,27].

Unfortunately, it is not feasible to prospectively study and derive new ultrasound-based growth standards for every given cohort. As a solution, several research groups have accepted that the shape of the Hadlock curve may be appropriate, representing the rate of growth of healthy fetuses across gestation, but that the expected mean birthweights should be adjusted (or weighted) to represent the birthweights of a local obstetric population [26,28]. The two most well-known examples of this are "A global reference for fetal weight and birthweight percentiles" by Mikolajzcyk and colleagues from Lancet 2011 [28], and the customized growth curves by Gardosi and colleagues that further adjust the fetal centile by maternal and fetal characteristics [26].

Customization, however, has significant drawbacks. Adjusting growth curves according to maternal height, weight or ethnicity may disadvantage high-risk groups [29–31]. Customized growth standards are also complex, change over time, have algorithms hidden by the institutes that derive them, and have a cost associated with their use that may preclude access in many settings [30]. The fact that the customized centile calculator varies across years, without the contributors to the variance always being clearly described, can make them difficult to use in research settings, or when auditing population changes over time.

Recognizing these limitations, Mikolajzcyk and colleagues provide a much simpler alternative – adjusting only for population mean birthweights [28]. However, they provide only a single set of centiles for each gestational week, which means a fetus can be attributed a different centile simply if measured at the beginning or end of a gestational week [32]. They, like many other ultrasound-based charts, are also unisex [23,33]. Adjusting for fetal sex improves the identification of small fetuses at high risk of perinatal mortality by creating an equal distribution of male and female infants classified as small for gestational age [34,35]. Ultrasound advances mean that sex-specific obstetric growth charts are a realistic option.

We propose a new obstetric growth chart, the Fetal Region-Specific Optimized Growth Standard (FROGS). FROGS overcomes the limitations of existing growth standards and incorporates international recommendations [18,27]. It is a transparent, ultrasound-based growth chart that adjusts for the mean of the population it represents, provides exact centiles for each gestational day, and allows the option to adjust for fetal sex (if known). We aimed to validate this novel growth standard in an Australian population using a large, statewide dataset, comparing to previously described, existing and widely used growth standards.

## Methods

### Part 1–Creation of the Fetal Region-specific Optimized Growth Standard (FROGS) chart

To create the FROGS chart, we adapted the published statistical methods from "A global reference for fetal-weight and birthweight percentiles" by Mikolajczyk and colleagues [28]. This chart was derived from the widely used and accepted Hadlock ultrasound-based weight standard [25]. Mikolajczyk and colleagues incorporated the 'proportionality approach' proposed by Gardosi and colleagues in their creation of customized centiles. The Hadlock curve shape is maintained, but kept in proportion to a local term mean birthweight, which can be adjusted [26].

This approach was chosen for several reasons. First, it is based on a fetal curve, aligning with international guidelines [18,27]. Second, given the widespread use of Hadlock's curve and its derivatives, we felt this ensured acceptability to a broad obstetric community. Finally, it allowed us to generate a simple, and completely transparent formula for future use by clinicians and researchers globally.

Hadlock's fetal weight formula is:

$$\text{Fetal weight (g)} = \exp(0.578 + 0.332 \times \text{gestational age (in weeks)} - 0.00354 \times \text{gestational age}^2).$$

The 'proportionality approach' adjusts Hadlock's formula by weighting it according to the mean birthweight of the population for which it is designed, using several key steps. First, the mean birthweight and standard deviation at 40 weeks'

gestation (from 40 + 0 to 40 + 6 inclusive) for a given population is identified. The chosen mean birthweight is then expressed as a proportion of 3705 g, which was the mean birthweight at the mid-point of 40 weeks' gestation in Hadlock's original cohort [28]. For example, if the term mean birthweight of a population is 3500 g, then the ratio is 3500/3705, which is 0.94, or 94%, of Hadlock's original weight.

Second, this calculated ratio is then maintained at all earlier gestations, i.e., at 30 weeks' gestation, the mean weight would be 94% of Hadlock's original mean weight at 30 weeks' gestation, and so on. Third, the standard deviation at 40 weeks' gestation in the population is used to generate percentiles, assuming a normal distribution of birthweights. As identified in Hadlock's study, the standard deviation at term is applicable at preterm gestations [25], so percentiles can be generated for each gestational day back to 20 weeks' gestation.

If fetal sex is known, then the mean birthweight at 40 weeks' gestation for males or females respectively is used, instead of the combined population mean. The FROGS formula therefore provides the option for adjustment of fetal sex.

In this study, we used the Australian mean birthweight and standard deviation. This was taken from the Australian Institute of Health and Welfare (AIHW), with means derived from all singleton births between 2004 and 2013 (2,801,405 infants) [36]. The mean birthweight among this population data for male infants born at 40 weeks' gestation (from 40 + 0 to 40 + 6 inclusive) was 3641 g, for female infants was 3504 g, and for all infants combined was 3573 g. The standard deviation at 40 weeks' gestation for both sexes combined was 421 g, which, as a percentage of birthweight, was 11.8%. This percentage was used in all cases, as it was representative of both male (standard deviation 429 g, 11.8%) and female (standard deviation 412 g, 11.8%) infants. For all earlier gestations, the standard deviation used to derive percentiles remained constant at 11.8% of the mean birthweight for any given gestational age.

To generate an 'estimate range' for the weight centile, FROGS provides an upper and lower centile value, assuming a ±10% measurement error [37–40]. While error may be smaller in certain cases, and larger in others (particularly small preterm [41] or large term infants [38,42]), 10% provides a reasonable value that can give an indication of error. To calculate the 'estimate range', the estimated fetal weight would be multiplied by 0.9 to find a 10% underestimation, and 1.1 to find a 10% overestimation. The FROGS formula is then applied to these upper and lower values, in the steps above. This can be presented, as below:

*"Assuming a 10% measurement error, the true estimated fetal weight centile would fall between the **xth** and **yth** percentile."*

Note that this estimate range does not indicate the probability of the estimated fetal weight matching the true fetal weight but is designed to assist clinicians in identifying (i) that the estimated fetal weight has inherent measurement error, and (ii) the range of the estimated centile may be wider or narrower, depending on where the estimated fetal weight lies on the bell curve.

A detailed technical approach can be found in S1 File.

The calculator can be accessed via https://github.com/centilechart/frogs.

## Part 2–Validation of the Fetal Region-specific Optimized Growth Standard

The International Federation of Gynaecology and Obstetrics (FIGO) recommends two methods of validating a new growth standard [27]. The first method is statistical validation – that is, finding the chart that best matches the distribution of weights within the local population. FIGO suggests that this is achieved by finding a chart that follows a normal distribution. Such a chart is centered on the 50th percentile and identifies approximately 10% of infants below the 10th percentile and 10% above the 90th centile. The second method of validating a new growth standard is outcome-based validation. This means finding a chart for which the diagnosis of small for gestational age infants most accurately predicts adverse outcomes associated with fetal growth restriction. We used both methods to validate our new growth standard, the FROGS.

**Population dataset used for validation of the FROGS.** The dataset used to validate the FROGS chart included all infants born in the state of Victoria, Australia, between 2009 and 2019. Data were obtained from the Consultative Council on Obstetric and Paediatric Mortality and Morbidity (CCOPMM), the central agency that collects data on obstetric and perinatal outcomes within the state. Data quality has been audited and validated in previous research [43,44].

A pre-specified statistical analysis plan specifying *a priori* the outcomes to be examined was developed before data cleaning and analysis commenced (S1 File). Singleton pregnancies from 24 + 0 to 42 + 6 weeks' gestation at delivery were included. Exclusion criteria included multiple pregnancy, congenital anomalies of any type, termination of pregnancy, or those with missing infant sex, uncertain gestational age, or missing or implausible birthweight. Implausible birthweights had been previously identified within the dataset after application of GROW centiles for prior research, which automatically fails to generate a centile for implausible birthweights [34,45].

Gestation was calculated based on the date of birth relative to the estimated due date, which incorporated first trimester ultrasound confirmation if undertaken. Obstetric and perinatal outcomes were based on data collected routinely by the attending midwives during pregnancy, birth, and the postnatal period. Stillbirths were defined as the death of an infant prior to birth. Neonatal death was defined as death within the first 28 days after a livebirth. Perinatal mortality included stillbirths and neonatal deaths combined. There were no missing primary outcome (stillbirth) data.

**Growth charts used for validation of the FROGS chart.** The FROGS, and four commonly used obstetric growth standards were applied to the Victorian statewide dataset for validation. First, we applied Australian birthweight centiles ("AIHW charts"). These were derived from the Australian Institute of Health and Welfare (AIHW) National Perinatal Statistics Unit [46] data from 2004 to 2013, including 2,801,405 infants. They are sex specific and provide centiles according to each completed week of gestation, from 20 to 44 weeks. The comparison with AIHW charts was chosen to investigate the differences seen between a postnatal (birthweight) chart and a fetal (ultrasound based) chart.

Second, we applied the original "Hadlock chart" – a centile generating equation based on intrauterine, ultrasound-derived data [47]. Hadlock represents one of the most used fetal weight standards, which is inbuilt into the software packages of many ultrasound machines in Australia and worldwide. The comparison with Hadlock charts was designed to investigate for differences between standards when adjustments for local population birthweight means were made.

Third, we applied "The global reference for fetal-weight and birthweight percentiles", by Mikolajczyk and colleagues (the "global chart") [28]. This chart followed similar principles to our FROGS chart, with the ability to adjust for local population birthweight means. However, the global chart assigns centiles according to gestational week and not according to exact gestational day and it doesn't allow adjustment for fetal sex. For this study, the local population birthweight mean used for both the FROGS chart and the global charts was the same. The comparison to global charts therefore investigated the differences observed when using a chart that was specific for gestation in days and that adjusted for fetal sex.

Finally, we compared to INTERGROWTH fetal growth charts. The INTERGROWTH charts are an international, prospective fetal growth standard derived from healthy women with minimal identified risk factors for fetal growth restriction [23,30]. The comparison to INTERGROWTH charts represented the differences when using a prescriptive chart derived from a multi-ethnic population.

### Step 1–Statistical validation of the chart

Prior to validation, baseline characteristics of the Victorian statewide population dataset were summarized by mean (standard deviation), median (25th–75th percentile) and number (%) according to type and distribution of the data. Small for gestational age populations were described. Significance level was two-sided, set at 0.05 and not adjusted for multiple comparisons. Statistical analysis was conducted using Stata Version 16 (StataCorp. 2019. Stata Statistical Software: Release 16.1. College Station, TX, USA).

The aim of the statistical validation of the charts was to (a) compare the overall distribution of the population birthweight centiles, as classified by each growth standard and (b) assess the proportion of infants born <3rd centile, <10th

centile, <50th, >90th centile and >97th centile by FROGS and all other charts. These centile cutoffs were used because they are clinically relevant thresholds which impact management [1,2]. We also assessed the distribution of birthweight centiles visually using a histogram and compared the mean, median and 25th–75th centile ranges. This approach aligned with recommendations by FIGO [18]. We were not able to plot the centiles according to a histogram for the AIHW or global charts, as they only provide outputs in centile brackets (such as 5–10th centile, 25th–50th centile, etc.), rather than exact centile figures (e.g., 5th, 6th, 7th centile).

Second, we classified all infants born according to which day of the gestational week they were born; "0" being born on the first day of the gestational week, i.e., 35 + 0 weeks (245 days), 36 + 0 weeks (252 days) etc., "1" being born on the second day of the gestational week, i.e., 35 + 1 weeks (246 days), 36 + 1 weeks (253 days), etc., and so on. Once classified, we assessed the distribution of infants determined to be small for gestational age according to the day of the gestational week (i.e., how many born on the first day of the gestational week were classified as <10th centile, how many on the second day, and so on). We compared the distributions of small for gestational age classification by charts that report fetal weight centiles per number of completed weeks of gestation, (AIHW and global charts), to those that assign fetal weight centile according to the exact number of days of gestation (FROGS, Hadlock and INTERGROWTH).

**Step 2–Outcome-based validation of the FROGS chart**

We then undertook the second part of the validation. We aimed to identify the chart by which a small for gestational age diagnosis best predicts adverse outcomes associated with fetal growth restriction. Non-overlapping populations (populations classified as small by one chart, but not another) were described. Adverse outcomes were reported as the point estimate with Wilson 95% confidence intervals.

Our primary outcome was the risk of stillbirth among the <10th centile infants as classified by each chart. We assessed this in two ways. First, we calculated the relative risk of stillbirth for those classified as small (birthweight <10th centile) by each chart, compared to those born >10th centile. Second, we compared the relative risk of stillbirth amongst infants classified as small for gestational age by one chart, but not another (non-overlapping populations). Using this second approach, we had four potential sets of comparisons.

1. **AIHW and FROGS comparison:** The risk of stillbirth amongst infants classified as small for gestational age by FROGS, but not AIHW charts, compared to the risk of stillbirth amongst infants classified as small for gestational age by AIHW charts, but not FROGS charts.

2. **Hadlock and FROGS comparison:** The risk of stillbirth amongst infants classified as small for gestational age by FROGS charts, but not Hadlock charts, compared to the risk of stillbirth amongst infants classified as small for gestational age by Hadlock charts, but not FROGS charts.

3. **Global and FROGS comparison:** The risk of stillbirth amongst infants classified as small for gestational age by FROGS charts, but not global charts, compared to the risk of stillbirth amongst infants classified as small for gestational age by global charts, but not FROGS charts.

4. **INTERGROWTH and FROGS comparison:** The risk of stillbirth amongst infants classified as small for gestational age by FROGS charts, but not INTERGROWTH charts, compared to the risk of stillbirth amongst infants classified as small for gestational age by INTERGROWTH charts, but not FROGS charts.

Secondary outcomes included combined perinatal mortality (stillbirth or neonatal death within 28 days post livebirth), low 5-min Apgar scores <7 or <4, admission to the Neonatal Intensive Care Unit (NICU), instrumental birth and emergency cesarean section rates. Additional secondary outcomes included 'suspected poor fetal growth' as an indication for induction of labor (a proxy for antenatal suspicion of growth restriction); and 'fetal distress' as a reason for instrumental birth

PLOS Medicine

or cesarean section (as this increases the likelihood of the operative birth being due to fetal growth restriction, with the hypoxic challenge of labor unmasking placental insufficiency). Complete case analysis was used.

### Ethics

Ethical approval for the project was obtained from the Mercy Health Human Research Ethics Committee (2021-029). As this was a retrospective cohort study using de-identified data, individual patient consent was not required.

## Results

### Study population

Between 2009 and 2019 there were 851,840 births in Victoria. After exclusions, there were 756,534 infants available for analysis (Fig 1). Overall, there were 1,913 stillbirths within our study cohort (0.25%). Baseline characteristics of the Victorian cohort are presented in S1 Table.

The mean birthweight at 40 weeks' gestation for all sexes in the Victorian population was 3580 g (compared to 3573 g for the national average), and the standard deviation of birthweight was 421 g, which was identical to the national average. Given the similarities between these figures and the national averages, it was appropriate to use the mean birthweight and standard deviation derived from the national dataset in our Victorian-only population.

### Statistical validation of the charts

First, we assessed the proportions of infants classified as <3rd and <10th centile. The 10th centile across gestation is presented in S1A–S1E Fig. All charts except INTERGROWTH classified close to 10% below the 10th centile (see Table 1). The FROGS chart classified 10.4% of all infants below the 10th centile, and 3.4% below the 3rd centile. AIHW charts classified 9.3% below the 10th centile, and 2.5% below the 3rd centile. Hadlock charts classified 11.1% below the 10th centile, and 2.8% below the 3rd centile. Global charts classified 10.9% below the 10th centile, and 3.6% below the 3rd

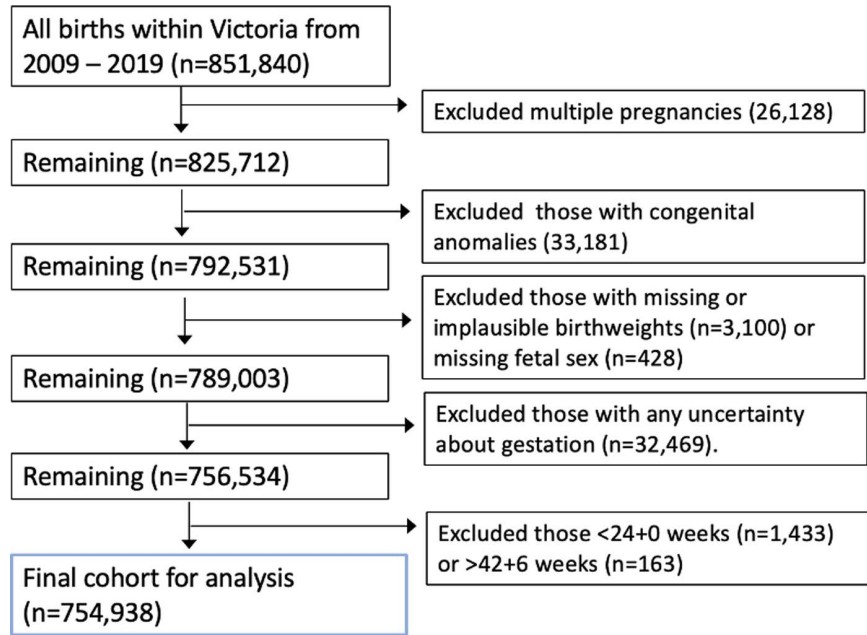

**Fig 1. Flow diagram of exclusions.**

centile. INTERGROWTH classified only 4.4% below the 10th centile, and 1.2% below the 3rd centile. For FROGS, AIHW and global charts (all adjusted for Australian population means), almost exactly half of all infants were classified below the 50th centile. However, Hadlock charts classified 59.8% below the 50th centile, and INTERGROWTH classified only 34.1% below the 50th centile.

We also assessed proportions of infants classified as >90th centile. AIHW classified 9.9% of all infants above the 90th centile, with 3.1% above the 97th centile (Table 1). Hadlock charts classified only 6.3% above the 90th centile. FROGS charts and global charts classified 13.2% and 13.7% above the 90th centile, respectively. For Hadlock and FROGS charts, this was largely due to an increased proportion of infants classified above the 97th centile (5.6% and 6.0%, respectively). INTERGROWTH classified 21.1% of the population as above the 90th centile, and 9.3% of the population above the 97th centile.

We then assessed the proportion of male and female infants classified as small for gestational age. Both FROGS and AIHW charts, which were sex-specific, had male infants representing half of the total small for gestational age population (Table 1). However, Hadlock, global and INTERGROWTH charts, which were not sex-specific, had an under representation of male infants in the total small for gestational age cohort (38.9%, 38.8% and 37.5% of the small for gestational age population being male, respectively).

The proportion of infants classified as small or large within different gestational age brackets (<28 weeks, 28–33 weeks, 34–36 weeks, ≥37 weeks) is reported in S2 Table. This demonstrates the failure of birthweight (AIHW) charts to identify the disproportionate number of preterm births that are small for gestational age.

Second, we assessed the distribution of the birthweights of infants classified according to the day of the gestational week on which they were born. We looked at charts that provided only centiles according to each completed gestational week ('week' charts; AIHW and global charts). We then examined charts that assigned centiles according to the exact gestational day ('day' charts, FROGS, Hadlock and INTERGROWTH charts). For the 'week' charts, the proportion of infants classified as small for gestational age reduced as the week progressed. Twice as many infants were classified as small for gestational age if born on the first day of the gestational week, as compared to the last (Fig 2A). For the day charts, there was an approximately equal distribution of infants classified as small for gestational age across the week (Fig 2B).

Finally, we assessed the distribution of centiles between Hadlock charts, FROGS charts and INTERGROWTH charts. These were the only charts that provided exact birthweight centiles (e.g., 26th centile), rather than birthweight centile brackets (e.g., 25–50th centile), which meant that these were the only distributions could be directly compared. A histogram comparing the distribution of the three charts (Fig 3) demonstrated a more even distribution of centiles using the FROGS centile, although FROGS still classified a greater proportion of infants as >95th centile than expected. Hadlock demonstrated a slight left skew, and INTERGROWTH a significant right skew.

**Table 1. Proportion of infants born <3rd, <10th, >90th and >97th centile by each obstetric growth standard, $n = 754,938$ total cohort.**

|  | FROGS | AIHW | Hadlock | Global | INTERGROWTH |
|---|---|---|---|---|---|
| <3rd centile | 25,331 (3.4%) | 18,642 (2.5%) | 21,139 (2.8%) | 26,957 (3.6%) | 8,922 (1.2%) |
| <10th centile | 78,789 (10.4%) | 70,119 (9.3%) | 84,126 (11.1%) | 82,169 (10.9%) | 33,507 (4.4%) |
| <50th centile | 372,754 (49.4%) | 377,071 (49.9%) | 451,489 (59.8%) | 374,145 (49.6%) | 257,320 (34.1%) |
| >90th centile | 99,818 (13.2%) | 75,054 (9.9%) | 47,425 (6.3%) | 103,496 (13.7%) | 159,274 (21.1%) |
| >97th centile | 42,613 (5.6%) | 23,492 (3.1%) | 15,529 (2.1%) | 45,195 (6.0%) | 70,582 (9.3%) |
| % male born <10th centile | 38,424 (48.8%) | 36,022 (51.4%) | 32,694 (38.9%) | 31,918 (38.8%) | 12,564 (37.5%) |

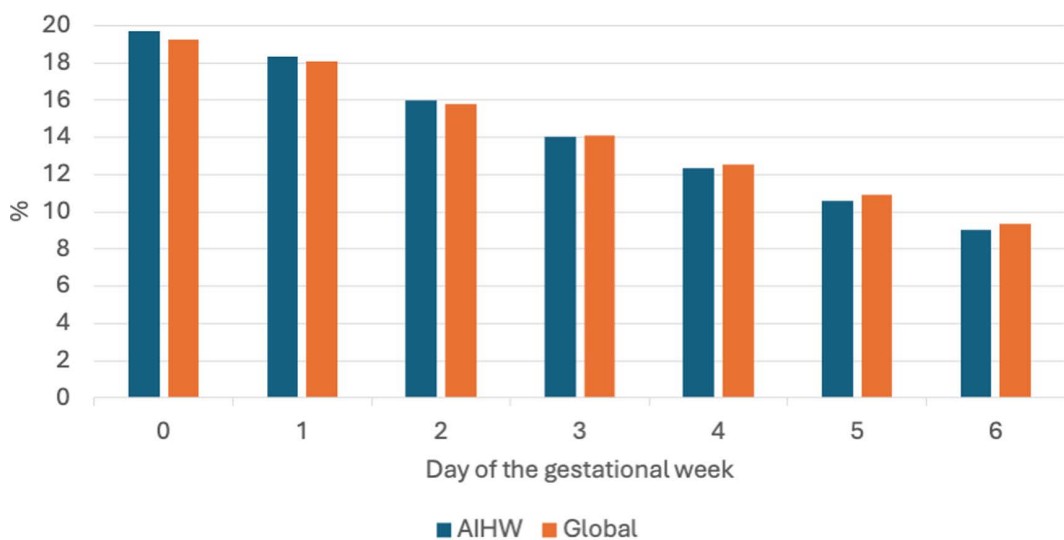

A) Proportion of infants classified as SGA by day of the gestational week - week charts

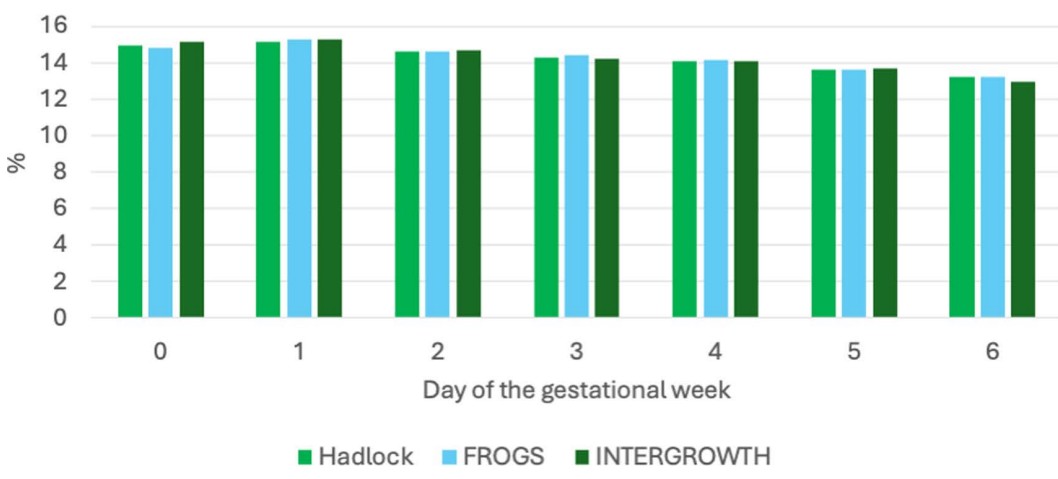

B) Proportion of infants classified as SGA by day of the gestational week - day charts

**Fig 2. Proportion of infants classified as SGA by day of the gestational week:** (A) Week charts, (B) Day charts.

## Outcome based validation of the FROGS chart

Amongst infants classified as small for gestational ages by each chart, the stillbirth rate per 1,000 births was 8.5 by AIHW, 10.2 by Hadlock, 10.5 by global, 11.0 by FROGS and 20.3 by INTERGROWTH. Comparatively, the risk of stillbirth for an infant not classified as small for gestational age by any chart, was 1.5 per 1,000 births.

The relative risk of stillbirth if an infant was SGA compared with non-SGA was 3.37 (95% CI [3.15, 3.60]; $p < 0.0001$) for AIHW charts, 6.51 (95% CI [5.95, 7.12]; $p < 0.0001$) for Hadlock charts, 6.69 (95% CI [6.11, 7.32]; $p < 0.0001$) for global charts, 7.05 (95% CI [6.45, 7.72]; $p < 0.0001$) for FROGS, and 11.9 (95% CI [10.9, 13.1]; $p < 0.0001$) for INTERGROWTH.

**AIHW compared with FROGS chart.** Of the 70,119 infants that were small for gestational age by AIHW charts, 5,335 (7.6%) were not classified as small for gestational age by FROGS. Of the 78,789 infants that were small for gestational age by FROGS charts, 14,005 (17.8%) were not classified as small for gestational age by AIHW (Fig 4A). Table 2 shows the obstetric and perinatal outcomes in these non-overlapping cohorts.

The population that was small for gestational age by FROGS only was at higher risk of all adverse outcomes, compared with those classified as small for gestational age by AIHW only. Those that were small for gestational age only by FROGS charts, but not AIHW charts, had a 13-fold higher risk of stillbirth (RR 13.14, 95% CI [6.51, 26.52], $p < 0.0001$). They had higher risks of perinatal death, admission to the NICU, and 5 min Apgars <7 and <4 (Table 2). They were also more likely to be born by operative delivery for fetal distress, and less likely to have had an unassisted vaginal birth.

**Hadlock compared with the FROGS chart.** Next, Hadlock charts were compared with the FROGS chart. Of the 73,379 infants that were small for gestational age by Hadlock charts, 11,064 (13.2%) were considered small for gestational age by Hadlock charts, but not by FROGS. Of the 78,789 infants that were small for gestational age by FROGS charts, 5,729 (7.3%) were not classified as small for gestational age by Hadlock charts (Fig 4B). Obstetric and perinatal outcomes in these non-overlapping cohorts were compared (Table 3).

FROGS identified an additional subpopulation of infants as small for gestational age that were at higher risk of experiencing adverse outcomes than the subpopulation identified as small for gestational age by Hadlock charts only. The small for gestational age by FROGS charts only cohort, compared to those classified as small by Hadlock charts only, were at double the risk of stillbirth (RR 2.14, 95% CI [1.2,3.56], $p = 0.0028$). Infants classified as small by FROGS only were at increased risk of perinatal death, NICU admission and 5-min Apgar scores <7 (Table 3). They were also at increased risk of being born by operative delivery, especially for fetal distress. Both groups were equally likely to have timed delivery for suspected poor fetal growth.

**Global compared with the FROGS chart.** Global charts were compared with the FROGS chart. These charts were the most similar, but the FROGS chart reports centiles by exact gestational day (not by completed gestational weeks) and adjusts for fetal sex. Of the 82,169 infants that were classified as small for gestational age by global charts,

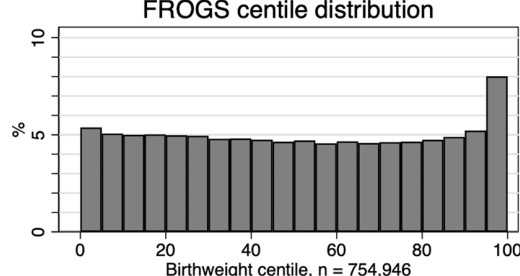

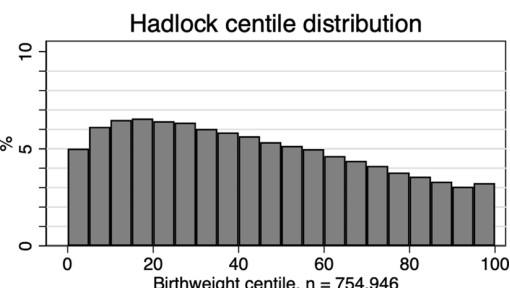

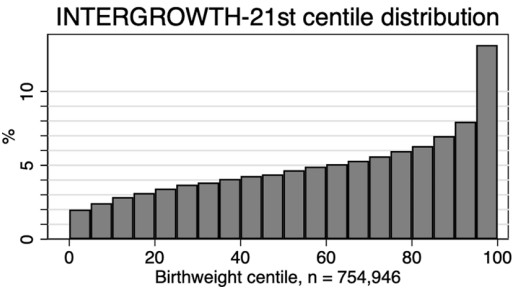

**Fig 3. Histogram of distribution of birthweight centiles by (A)** Custom centile and **(B)** Hadlock centile.

11,147 (13.6%) were not classified as small for gestational age by the FROGS. Of the 78,789 infants that were small for gestational age by FROGS, 7,767 (9.9%) were not classified as small for gestational age by the global charts ([Fig 4C]). Obstetric and perinatal outcomes in these non-overlapping cohorts were compared ([Table 4]).

The cohort of infants that were considered small for gestational age by FROGS only were compared to those considered small for gestational age by global charts only. They were found to be at increased risk of stillbirth (RR 1.54, 95% CI [1.00, 2.37], $p=0.0462$) and of 5 min Apgar scores <7 (RR 1.35, 95% CI [1.17, 1.57], $p=0.0001$). They were more likely to have had an operative delivery for fetal distress (RR 1.35, 95% CI [1.17, 1.57] ($p<0.0001$), and less likely to have had an unassisted vaginal birth ($p<0.0001$). They were not at significantly increased risk of perinatal death, NICU admission or

A)

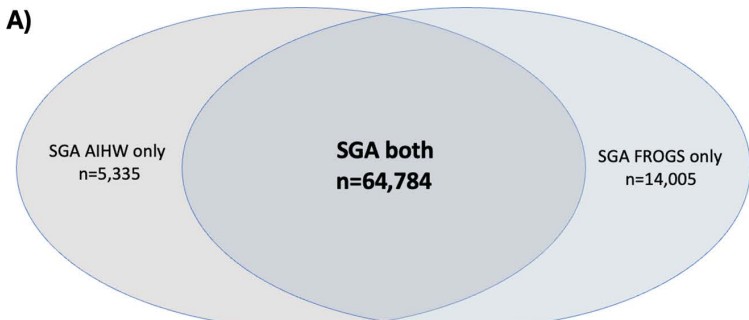

B)

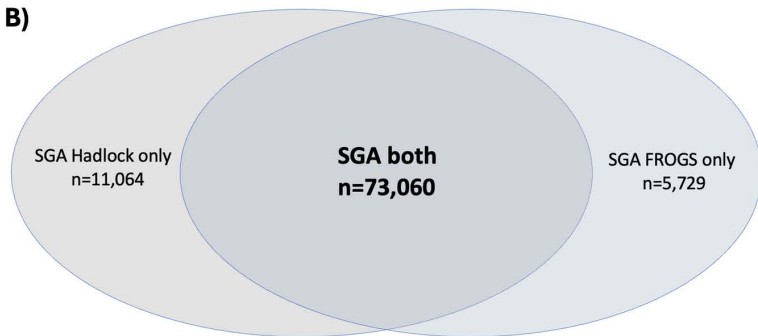

C)

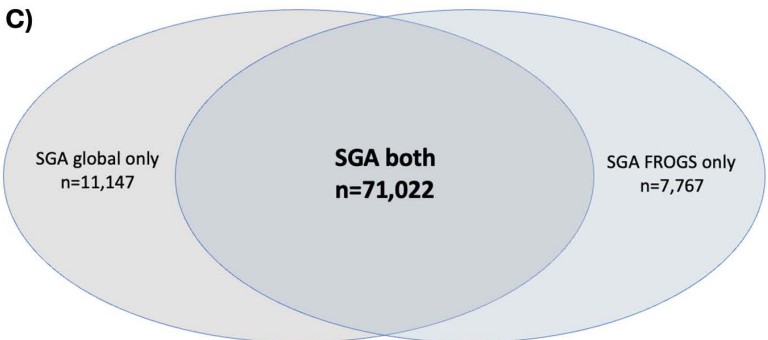

**Fig 4. Venn diagram showing non-overlapping populations for** (A) AIHW compared with FROGS charts, (B) Hadlock compared with FROGS charts, and (C) Global compared with FROGS charts.

**Table 2. Obstetric and perinatal outcomes in non-overlapping small for gestational age populations, by AIHW or FROGS charts*.**

| Outcome | Small for gestational age by AIHW charts but not small for gestational age by FROGS charts (5,335) *Reference group* | Small for gestational age by FROGS charts but not small for gestational age by AIHW charts (14,005) | Relative risk (95% confidence interval) |
|---|---|---|---|
| Stillbirth | 8 (0.15%) | 276 (2.0%) | 13.21 (6.51–26.52) $p < 0.0001$ |
| Perinatal death | 9 (0.17%) | 296 (2.1%) | 12.53 (6.46–24.30) $p < 0.0001$ |
| NICU admission | 30 (0.56%) | 1,185 (8.5%) | 78.2 (54.4–112.2) $p < 0.0001$ |
| Apgar score <7 at 5 min of life | 130 (2.4%) | 1,074 (7.8%) | 3.18 (2.66–3.81) $p < 0.0001$ |
| Apgar score <4 at 5 min of life | 19 (0.36%) | 253 (1.83%) | 5.13 (3.22–8.16) $p < 0.0001$ |
| Delivered for suspected poor growth | 454 (8.5%) | 2,325 (16.6%) | 1.95 (1.77–2.15) $p < 0.0001$ |
| Operative delivery for fetal distress | 915 (17.2%) | 2,838 (20.3%) | 1.18 (1.11–1.26) $p < 0.0001$ |
| Delivery mode | | | |
| Unassisted vaginal birth | 2,834 (53.1%) | 6,587 (47.0%) | $p < 0.0001$ |
| Instrumental birth | 907 (17.0%) | 1,749 (12.5%) | |
| Elective CS | 899 (16.9%) | 1,874 (13.4%) | |
| Emergency CS | 695 (13.0%) | 3,795 (27.1%) | |

*Note: AIHW charts are birthweight charts.

very low Apgar scores <4 at 5 min (Table 4). Both cohorts were equally likely to have had their delivery timed because of suspected poor growth.

**INTERGROWTH compared with the FROGS chart.** Finally, INTERGROWTH charts were compared with the FROGS charts. Of the 33,507 infants classified as small for gestational age by INTERGROWTH, only 54 (0.16%) were not classified by FROGS. Of the 78,789 infants that were small for gestational age by FROGS, 45,336 (57.5%) were not classified as small by INTERGROWTH charts.

Statistical comparisons between these groups were not performed, given the lack of power associated with such small numbers ($n = 54$). We noted that only 4.4% of the population was classified as <10th centile by INTERGROWTH, with a stillbirth risk per 1,000 of 20.3. Comparatively, FROGS classified 3.4% of the population as <3rd centile, with a stillbirth risk per 1,000 of 24.2.

## Discussion

### Main findings

We present the Fetal Region-specific Optimized Growth Standard (FROGS) – developed and validated following international guidelines. We applied FROGS to a statewide Australian dataset and compared its performance to four major, currently used, growth standards. FROGS more precisely identifies infants at high-risk of serious adverse perinatal complications, including stillbirth, compared with those charts that identified a similar proportion of infants as <10th centile. FROGS could improve pregnancy care and perinatal outcomes, given surveillance and timely delivery of at-risk pregnancies reduces stillbirth risk [48].

Our research confirms the importance of several features in an ideal growth standard. First, charts based on in utero ultrasound measurements, rather than birthweights, better identify preterm infants who experienced pathological growth [25,26]. This is consistent with international research and guidelines [18,27], and was evident when we compared the FROGS to the AIHW birthweight charts, which failed to identify many high-risk, preterm infants.

**Table 3. Obstetric and perinatal outcomes in non-overlapping small for gestational age populations, by Hadlock or FROGS charts.**

| Outcome | Small for gestational age by Hadlock charts but not small for gestational age by FROGS charts (11,064) *Reference group* | Small for gestational age by FROGS charts but not small for gestational age by Hadlock charts (5,730) | Relative risk (95% confidence interval) |
|---|---|---|---|
| Stillbirth | 28 (0.25%) | 32 (0.54%) | 2.14 (1.28–3.56) $p=0.0028$ |
| Perinatal death | 35 (0.32%) | 32 (0.56%) | 1.77 (1.09–2.85) $p=0.0182$ |
| NICU admission | 103 (0.93%) | 107 (1.87%) | 2.01 (1.53–2.62) $p<0.0001$ |
| Apgar[5] scores <7 at 5 min of life | 289 (2.62%) | 239 (4.20%) | 1.60 (1.35–1.89) $p<0.0001$ |
| Apgar[5] scores <4 at 5 min of life | 49 (0.44%) | 37 (0.65%) | 1.46 (0.96–2.24) $p=0.0787$ |
| Delivered for suspected poor growth | 889 (8.04%) | 501 (8.74%) | 1.09 (0.98–1.21) $p=0.114$ |
| Operative delivery for fetal distress | 1,872 (16.9%) | 1,287 (22.5%) | 1.33 (1.25–1.41) $p<0.0001$ |
| Delivery mode | | | |
| Unassisted vaginal birth | 6,210 (56.1%) | 2,895 (50.5%) | $p<0.0001$ |
| Instrumental birth | 1,848 (16.7%) | 1,080 (18.9%) | |
| Elective CS | 1,403 (12.7%) | 649 (11.3%) | |
| Emergency CS | 1,603 (14.5%) | 1,104 (19.3%) | |

**Table 4. Obstetric and perinatal outcomes in non-overlapping small for gestational age populations, by global or FROGS charts.**

| Outcome | Small for gestational age by global charts but not small for gestational age by FROGS charts (11,147) *Reference group* | Small for gestational age by FROGS charts but not small for gestational age by global charts (7,767) | Relative risk (95% confidence interval) |
|---|---|---|---|
| Stillbirth | 40 (0.36%) | 43 (0.57%) | 1.54 (1.00–2.37) $p=0.0347$ |
| Perinatal death | 53 (0.48%) | 44 (0.58%) | 1.19 (0.80–1.78) $p=0.3885$ |
| NICU admission | 207 (1.86%) | 154 (1.98%) | 1.07 (0.89–1.32) $p=0.5349$ |
| Apgar[5] scores <7 at 5 min of life | 352 (3.17%) | 332 (4.3%) | 1.35 (1.17–1.57) $p<0.0001$ |
| Apgar[5] scores <4 at 5 min of life | 52 (0.47%) | 50 (0.65%) | 1.38 (0.94–2.03) $p=0.1013$ |
| Delivered for suspected poor growth | 923 (8.3%) | 696 (9.0%) | 1.08 (0.99–1.19) $p=0.0997$ |
| Operative delivery for fetal distress | 1,804 (16.2%) | 1,739 (22.4%) | 1.38 (1.30–1.47) $p<0.0001$ |
| Delivery mode | | | |
| Unassisted vaginal birth | 6,164 (55.3%) | 3,945 (50.8%) | $p<0.0001$ |
| Instrumental birth | 1,780 (16.0%) | 1,432 (18.4%) | |
| Elective CS | 1,590 (14.5%) | 887 (11.4%) | |
| Emergency CS | 1,613 (14.5%) | 1,503 (19.4%) | |

Second, we confirm the importance of adjusting for local population means. When we assess distribution of centiles using the Hadlock chart – unadjusted for Australian mean birthweights – almost 60% of infants fall below the 50th centile. When we compared to INTERGROWTH, only 34.1% of infants fell below the 50th centile, and perhaps most notably, only 4.4% of infants fell below the 10th centile. This meant that the cohort of infants classified as small for gestational age by INTERGROWTH was, in size and risk, closer to the cohort classified as <3rd centile by other charts. Moreover, INTERGROWTH classified more than one in five infants above the 90th centile, and nearly one in 10 as >97th centile. This further limits interpretation and clinical applicability in our population. The importance of adjusting for local population means has previously been highlighted if <10th centile is to remain the threshold for obstetric interventions [18,27].

Third, our comparison with the global fetal and birthweight calculator confirms the importance of classifying centiles according to exact gestational day and fetal sex. Growth charts classifying centiles according to gestational week rather than day disproportionately classifies infants as small if measured early in the week [32]. Unisex growth standards underestimate the proportion of males – and overestimate the proportion of females – that are small and high-risk. Adjusting for fetal sex ensures an equal proportion of at-risk male and female infants are identified [34,49–53]. These features of FROGS ensure that infants classified as small are those at highest risk of pathological outcomes.

## Clinical implications

Most significantly, fetuses classified as small by FROGS are a cohort at high risk of stillbirth and thus represent the desired population to classify as small for gestational age. The introduction of this novel chart may enable clinicians to better identify fetuses at risk, offering surveillance and timely intervention to reduce this risk. The FROGS algorithm could be adapted to other local populations, if the population mean and standard deviation was known.

There has been considerable ongoing debate about which fetal growth standard should be used. Whether or not to customize centiles for maternal characteristics is potentially the most contentious issue [19,26]. Full customization based on maternal characteristics including maternal height, weight, parity and ethnicity may help identify infants at higher risk of adverse perinatal outcomes [54–56]. However, there has been significant controversy as to whether the marginal gains of full customization are worth the complexity of the algorithms, above an uncustomized population based chart [57,58]. Customization has other limitations. The algorithms used to derive customized growth standards are complex, change over time (where new chart updates are produced frequently), and the methods used to undertake the customization are not always disclosed by the institutes that generate them [30]. These factors can make customized growth standards difficult to apply clinically at a population level, or in research settings, as it can be hard to compare cohorts from different time points. Additionally, maternal height, weight and ethnicity are not purely physiological mediators of birthweight [59–64]. This has incited debate and hindered widespread adoption of customized charts [19,30,31,57,65]. Furthermore, in a superdiverse society like Australia, adjusting for ethnicity is not always feasible [66]. On a population level, we propose FROGS as a simple, transparent, stable weight centile. This centile provides a robust baseline for clinical management decisions based on fetal size, on which other local, and systemic, risk factors for adverse perinatal outcomes (for example, ethnicity) can be laid. The application, and impact, of such adjustments should be a focus of future research.

While customization may have benefits for individuals [26], in a population setting, we propose the FROGS as a simple, transparent, and acceptable middle ground. It encompasses the key features of growth standards known to have the greatest benefit.

'Prescriptive' growth standards – derived only from infants growing under optimal pregnancy conditions – are an alternative approach proposed for centile classification [67,68]. However, prescriptive charts may not be representative of population means [27,69,70]. Our findings are consistent with prior research that suggests INTERGROWTH charts classify markedly less than 10% as <10th centile [27,69,70]. Although different centile thresholds could be used, this would prove challenging in practice, where local and international clinical guidelines have widely recognized the 10th centile as threshold for increased surveillance and management [1,2,18,27]. Even if prescriptive charts are used that are representative of a given population, much of the benefit is simply seen with the use of a fetal, rather than birthweight chart [71]. There is limited additional gain from restriction to a healthy population [71].

As FROGS adjusts for population mean birthweights, it can represent the distribution of birthweights in any given population. This makes it practicable for use and translatable internationally. The FROGS algorithm is transparent, stable and the derived centiles can be updated if local mean birthweights change over time. While implementation of any clinical change always has pragmatic challenges, we believe that the ability of FROGS to generate a growth centile chart based on population-based birthweight data, one of the most widely accessible epidemiological data points, will limit the barriers to change.

Our findings suggest benefit in adjusting for fetal sex. However, fetal sex is not always known antenatally, and in some countries reporting the fetal sex prenatally is prohibited, to avoid sex-selective termination of pregnancy [72]. Using the sex-unadjusted option of the FROGS algorithm is a practical option in cases where sex is unknown but may reduce the potential benefits of the FROGS chart. There is also the possibility for fetal sex to be noted by the sonographer, but not disclosed to the parents. A similar practice can be seen in the context of non-invasive prenatal testing, where results are frequently known to the clinician but not disclosed to the patient.

## Strengths and limitations

The design of FROGS was based on international guidelines and principles used successfully when developing other growth standards [25,26]. Our large, population-based validation cohort enabled the examination of highly relevant clinical outcomes such as stillbirth. This demonstrates the potentially significant benefit of applying this novel standard clinically. We systematically validated the FROGS, using the methodology recommended by FIGO in a prospective, prespecified analysis. This two-step validation provides reassurance that the improvements seen when adopting the proposed growth standard may be generalizable. We compared FROGS to four other widely used growth charts that each differed in specific ways, enabling individual interrogation of the key features of FROGS. Classification as small by FROGS consistently outperformed the other charts in prediction of small infants at risk of stillbirth.

One of the strengths of our study is the comparison of the FROGS algorithm to other charts with shared features. This helped us draw conclusions about the specific attributes of FROGS with clinical benefit. For example, comparison to the original Hadlock formula highlighted the differences if local population birthweight and fetal sex are adjusted for, while comparison to a population-based birthweight chart reinforced the benefits of using a fetal-weight standard. While use of these comparison charts has helped us clarify these advantages, the global utility of FROGS could be further investigated by comparing to local growth standards derived using different methodology.

There are inevitable limitations in applying a growth standard to a large retrospective dataset of infants already born. All outcome measures were surrogate markers of fetal growth restriction. Furthermore, when examining stillbirth, the gestational age and birthweight at birth may not accurately reflect the exact gestational age and birthweight at the time of fetal death, which may introduce bias. However, we would expect this bias to be equal across all growth standards applied to our dataset. In addition, multiple growth standards were in clinical use in Victoria over the study timeframe. Fetuses that were identified as small on the chart in use would have been more likely to be identified antenatally and have risk mitigation strategies in place to reduce the risk of adverse outcomes. This may have affected our findings.

A limitation of the FROGS growth standard is that it works under several assumptions; that the shape of the Hadlock curve is reasonable, the birthweights are normally distributed, and that the standard deviation remains in a constant proportion to the mean at all earlier gestations. It is possible that these assumptions do not hold in all circumstances [73]. Prospective application of FROGS to an ultrasound population to test these assumptions is an important next step.

FROGS classified slightly more than expected >90th and >97th centile. FROGS used mean and standard deviation to identify the spread of percentiles, and this demonstrates that the upper extremes of birthweight are more widely distributed than the predicted distribution [73]. This overclassification could be seen as a limitation. However, this likely represents a genuine increase in the proportion of large infants in Australia [69], driven by increased maternal obesity [74–77] and diabetes [78,79], as has been previously reported. There could be concern regarding potential for over intervention in large infants. However, at the upper range of fetal size, absolute weight, rather than centile, may be a more relevant measure to trigger intervention [80–83] as size-related birth complications occur most commonly in the very largest infants [84].

## Research implications

The FROGS offers potential research and clinical applications, given the transparency of the formula, the double validation performed, and adjustment for leading contributors of centile misclassification (fetal sex, and day of gestation). Our

obstetric growth standard can be freely translated anywhere, by adjusting the mean birthweight and standard deviation. While in general a statistical model would be developed in one population and validated in an external population, the design of FROGS is such that adjusting for a local population mean and standard deviation is necessary. Further research should include validating the FROGS in external populations, using their local population birthweight mean and standard deviation to confirm international applicability.

We have developed and robustly validated a novel obstetric growth standard, the Fetal Region-specific Optimized Growth Standard (FROGS), following international recommendations. Adjustment for local population size distributions, fetal sex and exact gestational day resulted in a growth standard that was superior to existing, widely used charts in identifying small infants at risk of adverse perinatal outcomes. FROGS can be freely and easily adapted for global populations to enable implementation in varied clinical and research settings. Adoption of this free, novel growth standard could plausibly translate to meaningful improvements in obstetric care and perinatal outcomes.

## Supporting information

**S1 File. Statistical Analysis Plan.**
(DOCX)

**S1 STROBE Statement. STROBE statement.**
(DOCX)

**S1 Table. Baseline characteristics of cohort included in the analysis.**
(DOCX)

**S2 Table. Proportion of infants born <3rd, <10th, >90th and >97th centile by each obstetric growth standard, by gestation.**
(DOCX)

**S1 Fig. 10th centile across gestation.** (A) 24–28 weeks (male), (B) 29–34 weeks (male), (C) >34 weeks (male), (D) 24–28 weeks (female), (E) 29–34 weeks (female), and (F) >34 weeks (female).
(TIFF)

## Acknowledgments

We are grateful to the Consultative Council on Obstetric and Paediatric Mortality and Morbidity (CCOPMM) for providing access to the data used for this project and for the assistance of the staff at Safer Care Victoria. The conclusions, findings, opinions and views or recommendations expressed in this paper are strictly those of the author(s). They do not necessarily reflect those of CCOPMM.

## Author contributions

**Conceptualization:** Natasha L. Pritchard, Stephen Tong, Teresa MacDonald, Elizabeth McCarthy, Lisa Hui, Michael Permezel, Susan P. Walker, Anthea C. Lindquist.

**Data curation:** Natasha L. Pritchard.

**Formal analysis:** Natasha L. Pritchard, Hannah G. Gordon, Susan P. Walker.

**Funding acquisition:** Natasha L. Pritchard, Stephen Tong, Roxanne Hastie.

**Investigation:** Elizabeth McCarthy, Michael Permezel, Susan P. Walker, Anthea C. Lindquist.

**Methodology:** Natasha L. Pritchard, Elizabeth McCarthy, Michael Bethune, Emerson Keenan, Susan P. Walker, Anthea C. Lindquist.

Project administration: Roxanne Hastie, Anthea C. Lindquist.

Resources: Stephen Tong, Lisa Hui.

Software: Emerson Keenan.

Supervision: Stephen Tong, Teresa MacDonald, Lisa Hui, Roxanne Hastie, Susan P. Walker.

Validation: Teresa MacDonald, Michael Bethune, Hannah G. Gordon, Emerson Keenan.

Visualization: Michael Bethune.

Writing – original draft: Natasha L. Pritchard, Susan P. Walker, Anthea C. Lindquist.

Writing – review & editing: Stephen Tong, Teresa MacDonald, Elizabeth McCarthy, Lisa Hui, Michael Bethune, Hannah G. Gordon, Roxanne Hastie, Emerson Keenan, Michael Permezel, Susan P. Walker, Anthea C. Lindquist.

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
