## [Editor Report · Decision Letter 0]

Dear Dr Pritchard,

Thank you for submitting your manuscript entitled "The Fetal Region-specific Optimised Growth Standard (FROGS) – A fetal and birthweight centile calculator validated in a national population" for consideration by PLOS Medicine.

Your manuscript has now been evaluated by the PLOS Medicine editorial staff and I am writing to let you know that we would like to send your submission out for external peer review.

However, before we can send your manuscript to reviewers, we need you to complete your submission by providing the metadata that is required for full assessment. To this end, please login to Editorial Manager where you will find the paper in the 'Submissions Needing Revisions' folder on your homepage. Please click 'Revise Submission' from the Action Links and complete all additional questions in the submission questionnaire. Please ensure that the supplementary files are provided in your resubmission.

Please re-submit your manuscript within two working days, i.e. by Sep 13 2024 11:59PM.

Feel free to email me at lgaynor@plos.org if you have any queries relating to your submission.

Kind regards,

Louise Gaynor-Brook, MBBS PhD

Senior Editor

PLOS Medicine

---

## [Decision Letter · Decision Letter 1]

Dear Dr Pritchard,

Many thanks for submitting your manuscript "The Fetal Region-specific Optimised Growth Standard (FROGS) – A fetal and birthweight centile calculator validated in a national population" (PMEDICINE-D-24-02989R1) to PLOS Medicine. The paper has been reviewed by subject experts and a statistician; their comments are included below and can also be accessed here: [LINK]

As you will see, the reviewers appreciated your study, but also had some concerns. Reviewer #3 and #4 were concerned about some of the language used, and whether this adequately reflected your results. Reviewer #2 asked for comparisons of FROGS with additional birthweight charts, a point which the Academic Editor thought was important. After discussing the paper with the editorial team and an academic editor with relevant expertise, I'm pleased to invite you to revise the paper in response to the reviewers' comments. We plan to send the revised paper to some or all of the original reviewers, and we cannot provide any guarantees at this stage regarding publication.

We ask that you submit your revision by Feb 04 2025 11:59PM. However, if this deadline is not feasible, please contact me by email, and we can discuss a suitable alternative.

Don't hesitate to contact me directly with any questions (lgaynor@plos.org).

Best regards,

Suzanne

Suzanne de Bruijn, PhD

Associate Editor, PLOS Medicine

Sbruijn@plos.org

On behalf of,

Louise

Louise Gaynor-Brook, MBBS PhD

Senior Editor

PLOS Medicine

lgaynor@plos.org

Comments from the reviewers:

Reviewer #1: General comments:

This publication describes the comparison of four different charts for calculating birthweight centiles: a novel, customised fetal Hadlock chart which the authors have named FROGS (Hadlock adjusted for local mean term birthweight and newborn sex); a standard, un-customised fetal Hadlock chart; a national Australian birthweight AIHW chart; and a published Mikolazcyk chart.

Overall, the manuscript is clearly written and well-presented and the methods are adequately described. This study is interesting and deserves wide dissemination

The authors argue successfully in favour of the FROGS chart, demonstrating that - although all four charts have similar screen positive rate for SGA at around 10% - the FROGS chart can capture more effectively pregnancies resulting in stillbirth (primary outcome) and other secondary outcomes.

The authors conclude that the reasons why the FROGS chart performs better compared to the other three charts are:

1. Calculation of exact birthweight centile for each gestational day, as opposed to centile for completed week.

2. Adjustment of the curve for the local population mean term birthweight and standard deviation.

3. Adjustment of the curve for the infant sex.

In my opinion, conclusion 1 is not a particularly novel or newsworthy. Since the early 1990s, it is universally accepted that gestational age should always be expressed in exact days when calculating reference ranges and percentiles, rather than using truncated weeks. Nevertheless, the comparisons presented here in the manuscript, quantify this effect neatly and eloquently.

Conclusions 2 and 3 are interesting. The authors make a compelling argument that adjustment for local population mean birthweight and also infant sex customisation, will increase the capture of stillbirth, without significantly changing the screen positive rate.

The authors present their methods transparently. However, both in the abstract and the discussion the authors make frequently a statement of opinion that "the FROGS centile development algorithm is simple, transparent and does not change with time. This means it can be easily adapted for local populations…"

Although it is transparent, it is not simple to adapt prospectively in the delivery of prenatal care. It is also incorrect to state that it does not change with time. The fact that FROGS is adjusted for local birthweight means that it would need to change with time as mean birthweights change, either in different times or locations. The authors contradict their previous assertion in page 22 where they state "it can be updated easily if local mean birthweights change over time".

For a chart to improve prenatal care it should be used when reporting prenatal ultrasound and EFW centiles. Most institutions use a commercially available software (Viewpoint, Astraia etc) when reporting ultrasound examinations, so that the scan measurements are plotted in centile charts. The authors propose that every hospital or health board should be able to simply calculate their local population birthweight distributions after applying all sensible exclusions. Then, that they can modify the commercially available software. Or that they could design a bespoke electronic calculator from scratch. This is not really plausible. This is why the 1991 Hadlock chart remains in common use in 2024.

The authors make a compelling argument that customisation by fetal sex will improve the identification of SGA babies at risk of stillbirth. It is easy to do this when the baby is born where the sex is obvious. But the fetal sex is not universally recorded on prenatal ultrasound. This is not a mandatory part of prenatal care. In some parts of the world reporting the fetal sex prenatally is prohibited.

In my opinion, customising EFW reports for sex and adjusting for reference charts for local birthweight distributions is scientifically interesting but not easy or practical. The authors make repeated statements to describe this as a strength, but instead they should have addressed this as a limitation.

Specific comments:

Page 3 The statement "The FROGS centile development algorithm is simple, transparent and does not change with time. This means it can be easily adapted for local populations and applied to clinical and research settings globally"

This is a statement of opinion that is not supported by the facts presented here

Page 8 "The standard deviation at 40 weeks' gestation for both sexes combined was 421g, which, as a percentage of birthweight, was 11.8%. This was used in all cases,…"

Did the authors use a fixed standard deviation of 421g across every gestational week from 20 to 42 weeks; or did they use a standard deviation calculated at each gestational age as 11.8% of the mean birthweight for that gestational age?

Page 10 Please define what was an implausible birthweight.

Page 21 "the introduction of this easily adoptable chart…" The authors make this repeated claim throughout the discussion, but they do not explain how this is easy to adopt.

Page 22 "it can also be updated easily if local mean birthweights change with time"

The authors contradict their own earlier statement that this would not change with time. For an organisation to periodically check their population birthweight distributions there has to be academic infrastructure and support. Adapting local tools for reporting centiles is not that easy.

Page 24 "FROGS can be freely and easily adapted…"

The authors repeatedly state this apparent ease but they have not explained how this is easy.

Reviewer #2: Thank you for the opportunity to review this article by Pritchard et al. on a the Fetal Region-specific Optimised Growth Standard (FROGS), a fetal and birthweight centile calculator validated in a national population. The authors propose to adopt Hadlock's fetal weight formula and use a proportionality approach to "adjust" for the population-specific birthweight distribution. Mean term Australian birthweight was expressed as a proportion of Hadlock's mean birthweight. The calculated proportion/ratio is maintained at earlier gestations and standard deviations (expressed as a percentage of mean birthweight) were assumed to remain constant across GA. Adjustment for fetal sex is optionable. In the second part, the authors compared their chart to three widely adopted alternative charts in two ways: a "statistical validation", in which they compare the % of infants below several pre-defined thresholds, and an "outcome-based" validation in which the authors compare the ability of each chart to predict adverse outcomes associated with FGR, the primary outcome being 'stillbirth.

The article is an example of our continued struggle to define appropriate boundaries for normal fetal growth and - by extension - birthweight. Comparative studies are very difficult to interpret because of the many methodological differences between different charts. The main strength of this study is that it provides a simple tool which can easily adopted to other populations, provided that mean and SD of birthweight are available for this population. Another strength is that the charts were compared to two other charts with shared features (i.e., the incorporation of Hadlock's EFW formula), which minimizes the bias introduced by different methodologies, increasing the chance that observed differences are indeed attributable to differences in birthweight. The comparison to Hadlock illustrates the effect of adjustment for Australian mean birthweights and fetal sex; the comparison to the global fetal and birthweight calculator illustrates the effect of adjustment for fetal sex and the importance of classifying centiles according to each day.

I have a few major comments which I think should be addressed before publication may be considered:

1. The primary outcome is stillbirth, for which both gestational and birthweight may be unreliable. How do the authors think this influenced their results? The association with perinatal death is also determined mostly by stillbirths, and much less by neonatal deaths.

2. The overall mortality rate was 0.25%, whereas the stillbirth rates in all SGA groups were higher: 0.85% for the AIWH, 1.02% by the Hadlock birthweight chart, 1.05% by the global chart and 1.1% by FROGS. What was the overall RR of SGA according to each chart, compared to non-SGA infants? Looking at the non-overlapping SGA populations, only those 'SGA by FROGS but not AIHW' have an absolute mortality rate that is higher than the "base" stillbirth rate in SGA infants; in the other two comparisons this risk was lower than the base rate in SGA infants. On the other hand, the 'extra' SGA infants who were 'overlooked' by FROGS but identified by the global standard had a stillbirth risk higher than the base stillbirth risk. Are 'extra' SGA infants at increased risk compared to the non-SGA population? I.e., is there an actual benefit for the new charts compared to already existing charts? And is this risk clinically relevant? Not just in terms of relative risk, but absolute differences, which may be different across gestational age? There is substantial evidence that the discriminative power of all charts, birthweight or fetal weight, are poor, with AUCs around 0.60. These nuances are important to show.

4. The authors use a birthweight chart from AIHW and conclude that these "fail to identify the disproportionate number of preterm births that are small for gestational age". It is inherent to the design of a population-based birthweight chart to have low sensitivity for detecting infants with growth abnormalities, as such infants were all included. I wonder why the authors chose to use these birthweight charts, especially since there is an alternative Australian birthweight chart available (10.5694/mja2.50676) which aims to address this issue by excluding all births initiated by obstetric intervention, to minimize the influence of decisions to deliver SGA babies before term. The difference between the 10th percentile of this "prescriptive" chart and the AIHW chart is large, especially in preterm infants. The authors argue in their discussion that "prescriptive charts are not centered upon population birthweight means", but Hadlock's formula was also based on healthy women and in the Technical Appendix in the Supplemental material it is also stated that mean birthweight and SD should be derived from a population without risk factors for abnormal fetal growth.

5. The methodology is based on assumptions: e.g. that the shape of the Hadlock curve is appropriate, that the proportional effect of population characteristics is constant across gestational age, and that SD remains constant across GA. The assumptions were not tested and I think this requires attention in the discussion section. Simply stating that others have used a similar approach is not sufficient, and the fact that these results were published in a high-impact journal does not change this.

Minor comments:

1. How do the authors explain the disproportionately large number of infants with a birthweight >p95 in a population born between 2009-2019, when mean and SD of birthweight are from 2019 data? Both mean and SD are influenced by the inclusion of large infants? This should be discussed in more detail.

2. The Results section contains a lot of repetition of information that is also available in the tables/figures. Some results are already discussed or interpreted; this belongs in the Discussion section.

3. It would greatly help my understanding of the results if the different charts (i.e., the 10th percentiles) could be plotted in a graph, instead of (just) these Venn diagrams. This would help understand which infants are supposed to benefit from this new chart.

4. There is an error in the section that compares Hadlock with the FROGS chart, where it says that 874,126 infants that were classified..

5. Supplementary Table 1. should contain more information on the distribution of gestational age.

6. I am confused with regard to the 'confidence interval'. I was aware that 10% measurement error is generally accepted for estimations of fetal weight (which, for birthweight, would be unacceptable! Imagine an infant with a birthweight of 3500 grams, who weighs 3150 grams on one scale and 3850 gram on another!). "Assuming a 10% measurement error, the true estimated fetal weight centile would fall between the Xth and Yth percentile". I tried the calculator for 2 term male infants: 3550 grams at 38+6 weeks and 3720 grams at 41+3 weeks. X and Y were p41.2 and p95 for the former, and p15.4 and p75.0 for the latter. What message do I take from this?

Reviewer #3:

This paper is concerned with developing a birthweight calculator. The proposed model seems like a simple yet reasonable adjustment of previously developed calculators and the development of a SAP in advance of data cleaning and analysis is desirable. Specific comments are given below.

Please discuss the statistical aspects of the estimation process since the description on page 12 is somewhat vague. Are the data spread a bit too thin when looking at centiles? Was there an implicit bias-variance (sometimes called calibration-sharpness) trade off when trying to adjust to all those targets? How was this best assessed?

The performance of the FROGS model in the >95% percentile appears unexpectedly poor, please discuss the potential reasons for this (beyond the short discussion at the end of page 23) and how the model may be amended to correct this behaviour.

Please add a critical discussion of

1 the selection of the comparators, especially beyond the global chart compared to their current usage worldwide and

2 the training and test sets and how they relate to the comparators' performance

For example, the authors use Australian mean birthweight and standard deviation for the multiplicative adjustment and then test their model to Australian data, is this correct?

Typically, independent or external validation (i.e. against a country whose data have not been used) would be seen as a robust test of the model's performance.

At the very least, the presentation and discussion should reflect this practice as appropriate, especially since PLoS Medicine takes a global view to public health so the utility of FROGS outside Australia should be discussed fairly.

When focusing on the prediction of a particular target, say the population that is characterised as small for gestational age, there are two types of error that may occur, pertaining to sensitivity and specificity (or false positives/negatives). Essentially, one may have an interval that is too wide and always contain the small children, or having an interval that contains exclusively small children but may miss some. Is this a concern in this prediction task and if so should the reporting focus on both aspects?

Please follow standard scientific practice and use precise/rigorous language without overstating the study findings.

For example, a confidence interval refers to the frequentist quantification of parameter (and not model) uncertainty under repeated sampling. Accounting for arbitrary measurement error of a single factor does not yield confidence intervals. In some scientific areas this would be called perturbation analysis or one-way deterministic sensitivity analysis, but this "interval" may not be associated with a nominal coverage of any sort.

Similarly, exact centiles suggest lack of approximation of any sort, a practically impossible task in any estimate, especially by the deterministic model proposed in this paper.

I am a statistician and not a gynecologist but I understand that the exact conception day is typically unknown. If this is true, then the claim for accuracy by day within the gestational week is less relevant perhaps?

Reviewer #4: This paper addresses an important and controversial topic. I think the goal is to have a published paper to adopt this process in Australia. I think the paper needs to have slightly more nuanced conclusions and clarify the methods as outlined above.

Introduction

1. The 2nd half of this sentence that states: "iInternational [sic] guidelines also recommend that an obstetric growth curve represents the local population it is used in. (Ref 18, 27)" is not entirely technically precise. FIGO (Ref 18) concludes that "Alternatively, international standards for growth may be used with locally appropriate thresholds for risk interpretation" meaning that the Hadlock 1991 reference could be used if the cutoffs selected to define SGA or LGA are tailored to a local population. I do think this summary of FIGO recommendations in the Introduction should be edited to be technically precise.

2. The statement that "This instability makes these growth standards very difficult to apply in databases, research settings, or to use at a population level." Is somewhat true, but I would argue that they are not "very difficult". Grantz et al has published a heteroscedastic model to easily implement customized growth charts (A new method for customized fetal growth reference percentiles. PLoS One. 2023;18(3):e0282791. PMID: 36928064.) and Jason Gardosi's customization can be purchased for clinical use - albeit cost might be an issue. Also, "have algorithms hidden by the institutes that derive them" is somewhat true although the customization method has been published multiple times by Gardosi et al and can be easily implemented (as per the Grantz paper). In fact, many countries use customized fetal growth charts (McCowan ref). The main disadvantage of customization is that it that there is controversy as to whether it improves detection of fetuses at risk for perinatal morbidity and mortality (Zhang, Carberry) Suggest editing these sentences to be more technically precise.

Methods

3. The proportionality assumption was recently examined and questioned by Grantz et al (A new method for customized fetal growth reference percentiles. PLoS One. 2023;18(3):e0282791. PMID: 36928064.) This potential limitation should be mentioned in the discussion.

4. For part 1, the reference for the Hadlock EFW formula is incorrectly labeled. The formula presented is not to calculate EFW which would be a formula as published in the 1985 paper. The Hadlock 1991 paper, with the formula presented calculates "median fetal weight for each gestational week".

5. Also, the authors state "mean" throughout Part 1 but do they mean median or 50th percentile?

6. For the Gardosi method, the standard deviation is not assumed to be "constant" as stated; rather, it's the coefficient of variation that is assumed to be constant. This should be corrected.

7. It's not clear where the 3705 g for Hadlock was published. Per the Hadlock 1991 paper, the 50th percentile at 40 weeks was 3,619 g. Need to provide a reference.

8. I'm not sure this is a correct statement: "As identified in Hadlock's study, the standard deviation at term is applicable at preterm gestations." As I point out above, it's the coefficient of variation that is constant. The value of the SD depends on the mean EFW, so will most certainly vary depending on gestational age since fetuses are smaller on average preterm than at term.

9. Along these lines, it's not clear what was done with the SD of 421 g - do the authors mean they used this SD at every gestational age? If so, that would not be correct. Or perhaps they mean they used the same SD regardless of fetal sex. The methods need to be clarified.

10. I defer to a statistician as to whether the calculation of the confidence interval incorporating 10% measurement error is correct. I'm not familiar with this procedure. In the very least, they probably should not refer to it as a confidence interval.

11. Since the authors are following FIGO guidelines for evaluating FROG, I don't understand the rationale behind comparing to birthweight charts since these are inherently flawed and aren't recommended for clinical use when monitoring fetal weight. The rationale given has already been thoroughly established. I'm not opposed to this exercise, but in multiple places in the paper including the abstract it's concluded that FROG is better than "x, y, z charts" - I don't think it's informative to include birthweight charts in the final conclusions.

12. The classification of fetal weight centiles using the "completed weeks" method is going to cause problems inherent to this method. The authors know that and conclude accordingly. However, in clinical practice, my understanding is that ultrasound software in clinical systems interpolate between the weeks so that a day specific EFW percentile is technically reported. So, it's an important issue for researchers to be aware of, but less so for clinical practice. The authors might at least expand upon this distinction in the discussion.

13. It would be important to know how accurate and valid the data are by adding a sentence and a reference.

Results

14. Table 2- it should be added to the footnote that these are based on birthweight (as opposed to fetal weight.)

15. Table 2 - It's important to know which fetal growth chart tends to be used in clinical practice. It seems that part of the problem with comparing stillbirth between these 2 groups is that presumably the fetuses that were SGA by the AIHW and Hadlock charts (which are more like the un-customized fetal charts?) were more likely to have been identified antenatally, and therefore monitored more closely and/or delivered to prevent a stillbirth. If customized charts aren't used antenatally, then these fetuses that were SGA by FROG but not AIHW may have been more likely to remain unidentified. Can the authors comment on this potential bias in the Discussion?

16. These findings are supported by the fact that the FROGS and Global charts are more similar. (Although see my earlier comment about exact day versus completed weeks.)

17. Discussion

18. The clinical implications are overstated given the limitations outlined above. The authors do call for prospective evaluation; however, it should be noted that an RCT would be needed to compare the charts, which has already been conducted and did not find an improvement in customization. (Matias Vieira et al. The DESiGN cluster randomised trial)

Minor

19. Suggest avoiding using "this" as a subject of a sentence because it's often not clear to the reader what "this" is referring to. Would replace "this" with a specific noun or pronoun that clearly identifies what the authors are referring to, depending on the context of your sentence. Examples: "This is a barrier…"; "To overcome this…"; "As this is a healthier cohort"; "However, this is not…" in the Introduction, page 4-5 (multiple places) and in multiple other places throughout.

---

* Please upload any figures associated with your paper as individual TIF or EPS files with 300dpi resolution at resubmission; please read our figure guidelines for more information on our requirements: http://journals.plos.org/plosmedicine/s/figures. While revising your submission, please upload your figure files to the PACE digital diagnostic tool, https://pacev2.apexcovantage.com/. PACE helps ensure that figures meet PLOS requirements. To use PACE, you must first register as a user. Then, login and navigate to the UPLOAD tab, where you will find detailed instructions on how to use the tool. If you encounter any issues or have any questions when using PACE, please email us at PLOSMedicine@plos.org.

* [EDITOR: CHECK FINANCIAL DISCLOSURES, COI, DAS, AND ETHICS STATEMENTS AND INCLUDE ANY NECESSARY REQUESTS]

* Please ensure that the study is reported according to the [XXXX] guideline and include the completed [XXXX] checklist as Supporting Information. When completing the checklist, please use section and paragraph numbers, rather than page numbers. Please add the following statement, or similar, to the Methods: "This study is reported as per [XXXX] guideline (S1 Checklist)."

FIGURES AND TABLES

SUPPLEMENTARY MATERIAL

REFERENCES

[STUDY TYPE-SPECIFIC REQUESTS - DELETE SECTIONS AS NECESSARY]

RCTs [REFER TO RCT CHECKLIST AND MEETING NOTES FOR DETAILS TO ADD]

* PLOS Medicine requires that all trials be prospectively registered in one of registries recognized by WHO. Please ensure that study registration details are included in the Methods section.

* Please structure the Methods section using the following sub-headings: Study design and participants, Randomization and masking, Procedures, Outcomes, Statistical analysis.

* The following outcomes measures [ADD DETAILS AS NEEDED OR DELETE BULLET POINT] appear to differ between the submitted manuscript and the protocol [and/or trial registry]. Please clarify and explain all discrepancies between the paper and protocol. If the outcomes were not prespecified in the protocol, please define them in the Methods (Outcomes section) as post hoc and explain why they were added. Post-hoc comparisons should be presented as hypothesis generating rather than conclusive.

* Please ensure that all prespecified outcomes (primary, secondary, and exploratory) are listed in the Methods/Outcomes section and indicate whether there are outcomes that are not presented in the current report.

* Please specify the dates (Month Day, Year) during which study enrollment and follow up occurred.

* Please include absolute numbers wherever you report percentages; eg, n/N (%)

* Please present the safety data for the study including numbers of specific events and whether or not adverse events are thought to be related to treatment. AEs should be reported in the abstract, per CONSORT and CONSORT-Harms.

* Please complete the CONSORT checklist (https://www.equator-network.org/reporting-guidelines/consort/) and ensure that all components of CONSORT are present in the manuscript, including how randomization was performed, allocation concealment, blinding of intervention, definition of lost to follow-up, power statement. When completing the checklist, please use section and paragraph numbers, rather than page numbers.

* Please report your abstract according to CONSORT for abstracts, following the PLOS Medicine abstract structure (Background, Methods and Findings, Conclusions) https://www.equator-network.org/reporting-guidelines/consort-abstracts/

* If your trial had to undergo important modifications in response to extenuating circumstances, please complete the CONSERVE-CONSORT checklist and provide in your Supporting Information; (https://www.equator-network.org/reporting-guidelines/guidelines-for-reporting-trial-protocols-and-completed-trials-modified-due-to-the-covid-19-pandemic-and-other-extenuating-circumstances-the-conserve-2021-statement/). When completing the checklist, please use section and paragraph numbers, rather than page numbers.

* In keeping with our commitment to Open Science, please include the study protocol document and analysis plan (including any amendments) as Supporting Information to be published with the manuscript if accepted.

* Please note that PLOS Medicine requires prospective, public registration of a data sharing plan (as part of mandatory clinical trials registration) for all clinical trials that began enrollment on or after January 1, 2019, in accordance with ICMJE requirements.

OBSERVATIONAL STUDIES

* Abstract: Please include the study design, population and setting, number of participants, years during which the study took place (enrollment and follow up), length of follow up, and main outcome measures.

* Please ensure that the study is reported according to the STROBE (or appropriate STOBE extension) guideline (available from: https://www.equator-network.org/reporting-guidelines/strobe) and include the completed STROBE (or STROBE extension) checklist as Supporting Information. Please add the following statement, or similar, to the Methods: "This study is reported as per the Strengthening the Reporting of Observational Studies in Epidemiology (STROBE) guideline (S1 Checklist)." When completing the checklist, please use section and paragraph numbers, rather than page numbers.

* [FOR POPULATION HEALTH/REGISTRY STUDIES] Please ensure that the study is reported according to the RECORD guideline (available from https://www.record-statement.org) and include the completed checklist as Supporting Information. Please add the following statement, or similar, to the Methods: "This study is reported as per the Reporting of Studies Conducted using Observational Routinely-Collected Data (RECORD) guideline (S1 Checklist)." When completing the checklist, please use section and paragraph numbers, rather than page numbers.

* [FOR POPULATION HEALTH ESTIMATES] Please ensure that the study is reported according to the GATHER statement (available from https://www.equator-network.org/reporting-guidelines/gather-statement) and include the completed checklist as Supporting Information. Please add the following statement, or similar, to the Methods: "This study is reported as per the Guidelines for Accurate and Transparent Health Estimates Reporting (GATHER) statement (S1 Checklist)." When completing the checklist, please use section and paragraph numbers, rather than page numbers.

* [FOR MEDIATION ANALYSES] We recommend that the study is reported according to the AGReMA statement (https://agrema-statement.org/#:~:text=AGReMA%20is%20an%20evidence%2D%20and,randomised%20trials%20and%20observational%20studies) and include the completed checklist as Supporting Information. Please add the following statement, or similar, to the Methods: "This study is reported as per the Guideline for Reporting Mediation Analyses (AGReMA) statement (S1 Checklist)." When completing the checklist, please use section and paragraph numbers, rather than page numbers.

* For all observational studies, in the manuscript text, please indicate: (1) the specific hypotheses you intended to test, (2) the analytical methods by which you planned to test them, (3) the analyses you actually performed, and (4) when reported analyses differ from those that were planned, transparent explanations for differences that affect the reliability of the study's results. If a reported analysis was performed based on an interesting but unanticipated pattern in the data, please be clear that the analysis was data driven.

* Please state in the Methods section whether the study had a prospective protocol or analysis plan. If a prospective analysis plan (from your funding proposal, IRB or other ethics committee submission, study protocol, or other planning document written before analyzing the data) was used in designing the study, please include the relevant document(s) with your revised manuscript as a Supporting Information file to be published alongside your study and cite it in the Methods section. A legend for this file should be included at the end of your manuscript. If no such document exists, please make sure that the Methods section transparently describes when analyses were planned, and when/why any data-driven changes to analyses took place. Changes in the analysis, including those made in response to peer review comments, should be identified as such in the Methods section of the paper, with rationale.

MODELLING STUDIES

The following list is derived from Geoffrey P Garnett, Simon Cousens, Timothy B Hallett, Richard Steketee, Neff Walker. Mathematical models in the evaluation of health programmes. (2011) Lancet DOI:10.1016/S0140-6736(10)61505-X:

* If pertinent, please provide a diagram that shows the model structure, including how the natural history of the disease is represented, the process and determinants of disease acquisition, and how the putative intervention could affect the system.

* Please provide a complete list of model parameters, including clear and precise descriptions of the meaning of each parameter, together with the values or ranges for each, with justification or the primary source cited and important caveats about the use of these values noted.

* Please provide a clear statement about how the model was fitted to the data, including goodness-of-fit measure, the numerical algorithm used, which parameter varied, constraints imposed on parameter values, and starting conditions.

* For uncertainty analyses, please state the sources of uncertainties quantified and not quantified [can include parameter, data, and model structure].

* Please provide sensitivity analyses to identify which parameter values are most important in the model. Uncertainty estimates seek to derive a range of credible results on the basis of an exploration of the range of reasonable parameter values. The choice of method should be presented and justified.

* Please discuss the scientific rationale for the choice of model structure and identify points where this choice could influence conclusions drawn. Please also describe the strength of the scientific basis underlying the key model assumptions.

* For studies that develop a prediction model or evaluate its performance, please ensure that the study is reported according to the TRIPOD statement (https://www.equator-network.org/reporting-guidelines/tripod-statement) and include the completed checklist as Supporting Information. Please add the following statement, or similar, to the Methods: "This study is reported as per the Transparent Reporting of a Multivariable Prediction Model for Individual Prognosis Or Diagnosis (TRIPOD) statement (S1 Checklist)." For studies using machine learning, please use the TRIPOD-AI checklist. When completing the checklist, please use section and paragraph numbers, rather than page numbers.

DIAGNOSTIC STUDIES

* Please ensure that the study is reported according to the STARD guideline (https://www.equator-network.org/reporting-guidelines/stard/) and include the completed STARD checklist as Supporting Information. Please add the following statement, or similar, to the Methods: "This study is reported as per the Standards for Reporting of Diagnostic Accuracy (STARD) guideline (S1 Checklist)." When completing the checklist, please use section and paragraph numbers, rather than page numbers.

* Please structure your Abstract according to STARD for Abstracts (https://www.equator-network.org/reporting-guidelines/stard-abstracts/).

* Please structure the Methods section using the following sub-headings: Study design, Participants, Test methods, Analysis.

* Please include a diagram to describe the flow of participants through the study (typically figure 1).

MENDELIAN RANDOMIZATION STUDIES

* Please ensure that the study is reported according to the STROBE-MR guideline (https://www.equator-network.org/reporting-guidelines/strobe/) and include the completed STROBE-MR checklist as Supporting Information. Please add the following statement, or similar, to the Methods: "This study is reported as per the Strengthening the Reporting of Observational Studies in Epidemiology (STROBE) guideline, specific for mendelian randomization (S1 Checklist)." When completing the checklist, please use section and paragraph numbers, rather than page numbers.

* In the Introduction, please describe the exposure and the evidence for a potential causal relationship between exposure and outcome.

* In the Methods, please explicitly state the 3 core instrumental variable assumptions for the main analysis (relevance, independence, and exclusion restriction), as well assumptions for any additional or sensitivity analysis.

* In the Methods, please describe the MR estimator (e.g., 2-stage least squares, Wald ratio) and related statistics. Detail the included covariates and, in case of 2-sample MR, whether the same covariate set was used for adjustment in the 2 samples.

* If you are presenting an instrumental variable estimate, please compare this to the conventional observational estimate.

* Report the associations between genetic variant and exposure and between genetic variant and outcome, preferably on an interpretable scale.

* Report MR estimates of the relationship between exposure and outcome and the measures of uncertainty from the MR analysis, on an interpretable scale, such as odds ratio or relative risk per SD difference.

* If relevant, please consider translating estimates of relative risk into absolute risk for a meaningful time period.

* Please consider including plots to visualize results (e.g., forest plot, scatterplot of associations between genetic variants and outcome vs between genetic variants and exposure).

SURVEY-BASED STUDIES

* Please ensure that the study is reported according to the CROSS guideline (https://www.equator-network.org/reporting-guidelines/a-consensus-based-checklist-for-reporting-of-survey-studies-cross/) and include the completed CROSS checklist as Supporting Information. Please add the following statement, or similar, to the Methods: "This study is reported as per A Consensus-Based Checklist for Reporting of Survey Studies (CROSS) guideline (S1 Checklist)." When completing the checklist, please use section and paragraph numbers, rather than page numbers.

* Please report your survey response rates according to AAPOR recommendations (https://aapor.org/standards-and-ethics/best-practices/)

* Please define how the population surveyed was sampled.

* Please compare characteristics of respondents and nonrespondents if possible.

* If sequential waves of the survey were sent, please specify whether the characteristics of respondents changed over time or remained constant.

* Please include the survey response rate in the Abstract.

* Please include a copy of the survey in the supplementary files.

SYSTEMATIC REVIEWS & META-ANALYSES

* Please report your SR/MA according to the PRISMA guidelines provided at the EQUATOR site. http://www.equator-network.org/reporting-guidelines/prisma/. Please provide the completed PRISMA checklist as Supporting Information. When completing the checklist, please use section and paragraph numbers, rather than page numbers. Please add the following statement, or similar, to the Methods: "This study is reported as per the Preferred Reporting Items for Systematic Reviews and Meta-Analyses (PRISMA) guideline (S1 Checklist)."

* Abstract: Please report your abstract according to PRISMA for abstracts (https://doi.org/10.1371/journal.pmed.1001419) following the PLOS Medicine abstract structure (Background, Methods and Findings, Conclusions). Please ensure you provide dates of search, data sources, number of studies included, types of study designs included, eligibility criteria, and synthesis/appraisal methods.

* Please note that we expect searches to be updated to within 6 months of the time of submission.

QUALITATIVE STUDIES

* Please report your qualitative study according to the appropriate study design provided at (http://www.equator-network.org/?post_type=eq_guidelines&eq_guidelines_study_design=qualitative-research&eq_guidelines_clinical_specialty=0&eq_guidelines_report_section=0&s=) and provide the relevant completed checklist as a supplemental file. In the checklist, please include sufficient text excerpted from the manuscript to explain how you accomplished all applicable items. When completing checklists, please use section and paragraph numbers, rather than page numbers.

* We recommend that authors use the COREQ checklist, or other relevant checklists listed by the Equator Network, such as the SRQR, to ensure complete reporting (see: http://www.equator-network.org/?post_type=eq_guidelines&eq_guidelines_study_design=qualitative-research&eq_guidelines_clinical_specialty=0&eq_guidelines_report_section=0&s=). Please add the following statement, or similar, to the Methods: "This study is reported as per the Consolidated criteria for reporting qualitative research (COREQ): a 32-item checklist for interviews and focus groups (S1 Checklist)."

* In general, we expect qualitative studies to include the following: 1) defined objectives or research questions; 2) description of the sampling strategy, including rationale for the recruitment method, participant inclusion/exclusion criteria and the number of participants recruited; 3) detailed reporting of the data collection procedures; 4) data analysis procedures described in sufficient detail to enable replication; 5) a discussion of potential sources of bias; and 6) a discussion of limitations.

HEALTH ECONOMICS / COST-EFFECTIVENESS STUDIES

* Please ensure that the study is reported according to the CHEERS guideline (available from: https://www.equator-network.org/reporting-guidelines/cheers) and include the completed checklist as Supporting Information. Please add the following statement, or similar, to the Methods: "This study is reported as per the Strengthening the Consolidated Health Economic Evaluation Reporting Standards 2022 (CHEERS 2022) Statement (S1 Checklist)." When completing the checklist, please use section and paragraph numbers, rather than page numbers.

---

## [Decision Letter · Decision Letter 2]

Dear Dr. Pritchard,

Thank you very much for re-submitting your manuscript "The Fetal Region-specific Optimised Growth Standard (FROGS) – A fetal and birthweight centile calculator validated in a national population" (PMEDICINE-D-24-02989R2) for review by PLOS Medicine.

I have discussed the paper with my colleagues and the academic editor and it was also seen again by 3 reviewers. As you can see below, the reviewers are satisfied, but there is one remaining request from the academic editor which we would like you to address. The academic editor asks for comparisons with the fetal weight charts Intergrowth and WHO. We would like you to include these comparisons, or provide explanations why these comparisons cannot be made. Furthermore, we would like you to clearly incorporate these comparisons in the manuscript. Provided this, and remaining editorial and production issues are dealt with, I am pleased to say that we are planning to accept the paper for publication in the journal.

[LINK]

We look forward to receiving the revised manuscript by Apr 07 2025 11:59PM.   

Sincerely,

Suzanne De Bruijn, PhD

associate Editor 

PLOS Medicine

Sbruijn@plos.org

***Requests from Editors:

Abstract: ‘background’ should be the rationale for the study, rather than the goal.

-Author summary: Last bullet point in ‘why’ should actually be in ‘what’

-legends: They’re all there, but some don’t provide enough information (e.g. , don’t mention what is in different panels), whereas others have a legend for each subpanel separately.

-p18: you mention ‘supplementary table 2s’, please remove the ‘s’.

GENERAL EDITORIAL REQUESTS

* At this stage, we ask that you include a short, non-technical Author Summary of your research to make findings accessible to a wide audience that includes both scientists and non-scientists. The Author Summary should immediately follow the Abstract in your revised manuscript. This text is subject to editorial change and should be distinct from the scientific abstract. Ideally each sub-heading should contain 2-3 single sentence, concise bullet points containing the most salient points from your study. In the final bullet point of ‘What Do These Findings Mean?’ Please include the main limitations of the study in non-technical language.

Please see our author guidelines for more information: https://journals.plos.org/plosmedicine/s/revising-your-manuscript#loc-author-summary."

* Please confirm that your title complies with to PLOS Medicine's style. Your title must be nondeclarative and not a question. It should begin with main concept if possible. "Effect of" should be used only if causality can be inferred, i.e., for an RCT. Please place the study design ("A randomized controlled trial," "A retrospective study," "A modelling study," etc.) in the subtitle (ie, after a colon).

* Please confirm that your abstract complies with our requirements, including providing all the information relevant to this study type https://journals.plos.org/plosmedicine/s/submission-guidelines#loc-abstract

* Please ensure that the Introduction ends with a clear description of the study question or hypothesis.

* Please ensure that all abbreviations are defined at first use throughout the text.

* Please confirm that all numbers presented in the abstract are present and identical to numbers presented in the main manuscript text.

GENERAL

* Please remove the 'conclusions' subheading.

* Statistical reporting: Please revise throughout the manuscript, including tables and figures.

- Please report statistical information as follows to improve clarity for the reader ""22% (95% CI [13%,28%]; p</=)"".

- Please separate upper and lower bounds with commas instead of hyphens as the latter can be confused with reporting of negative values.

- Please repeat statistical definitions (HR, CI etc.) for each set of parentheses."

* In the author summary, in the final bullet point of 'What Do These Findings Mean?', please include the main limitations of the study in non-technical language.

FUNDING STATEMENT

* The funding statement should include: specific grant numbers, initials of authors who received each award, URLs to sponsors’ websites. Also, please state whether any sponsors or funders (other than the named authors) played any role in study design, data collection and analysis, the decision to publish, or preparation of the manuscript. If they had no role in the research, include this sentence: “The funders had no role in study design, data collection and analysis, decision to publish, or preparation of the manuscript.”

COMPETING INTERESTS STATEMENT

* All authors must declare their relevant competing interests per the PLOS policy, which can be seen here: https://journals.plos.org/plosmedicine/s/competing-interests For authors with ties to industry, please indicate whether any of the interests has a financial stake in the results of the current study.

* Please add this statement to the manuscript's Competing Interests: "[Initials] is a paid statistical consultant on PLOS Medicine's statistical board."

DATA AVAILABILITY

* Please include the statement on code availability in the data availability statement.

FIGURES

* Please show graph axes beginning at zero. If this is not possible, please show a break in the axis.

***Request from the academic editor:

-Please compare FROGS to the fetal weight charts Intergrowth and WHO. As these are both used internationally and have online calculators, it would be good to provide a comparison to these fetal charts. If this is not possible, please provide some explanation why this has not been done.

other comment:

the discussion doesn't really tease out the thorny issue that adjusting for population means in a diverse population doesn't account for different rates of pathologically small babies in the different subgroup populations. I accept that FROGS is being presented as a middle ground but I think the point that adjusting for a local population where one group will much more commonly be <10th centile than another group still leaves the issue that you overtreat one group whilst under treating another. FROGS gets you nearer to you local population but doesn't actually address this challenge at all. Clearly further research of adverse outcomes in subgroups of a population (e.g. for our local population it would be a subgroup adjustment amongst our south asian women) is needed to determine whether a further adjustment for the local population should be applied or not.

***Comments from Reviewers:

Reviewer #1: This publication describes the comparison of four different charts for calculating birthweight centiles: a novel, customised fetal Hadlock chart which the authors have named FROGS (Hadlock adjusted for local mean term birthweight and newborn sex); a standard, un-customised fetal Hadlock chart; a national Australian birthweight AIHW chart; and a published Mikolazcyk chart.

Overall, the manuscript is clearly written and well-presented and the methods are adequately described. This study is interesting and deserves wide dissemination

The authors argue successfully in favour of the FROGS chart, demonstrating that - although all four charts have similar screen positive rate for SGA at around 10% - the FROGS chart can capture more effectively pregnancies resulting in stillbirth (primary outcome) and other secondary outcomes.

The authors conclude that the reasons why the FROGS chart performs better compared to the other three charts are:

1. Calculation of exact birthweight centile for each gestational day, as opposed to centile for completed week.

2. Adjustment of the curve for the local population mean term birthweight and standard deviation.

3. Adjustment of the curve for the infant sex.

In my opinion, conclusion 1 is not a particularly novel or newsworthy. Since the early 1990s, it is universally accepted that gestational age should always be expressed in exact days when calculating reference ranges and percentiles, rather than using truncated weeks. Nevertheless, the comparisons presented here in the manuscript, quantify this effect neatly and eloquently.

Conclusions 2 and 3 are interesting. The authors make a compelling argument for the adjustment of the Hadlock standard for local population mean birthweight and standard deviation; and also customisation for infant sex. The proposed FRGOS chart describes more accurately SGA infants at preterm gestations. The FROGS chart also increases the representation of stillbirth and other adverse perinatal outcome in the SGA category, without significantly changing the SGA screen positive rate.

The authors present their methods transparently.

In this revised version of their manuscript they have addressed previous reviewer comments.

Reviewer #2: Thank you for the opportunity to review this revised version of the article. I am pleased with the comprehensive responses the authors provided to my questions and the alterations that were made to the manuscript.

If case the article is accepted for publication, I hope the authors opt for publication of the peer review history because it provides a deepened and nuanced understanding of the article.

I have one minor comment regarding the discussion section: the author mention a "strength"in the section on research implications, this should be put in the strengths and limitations section.

Reviewer #3: This paper has been revised and my comments have been largely addressed. As such it appears acceptable for publication

[LINK]

---

## [Editor Report · Decision Letter 3]

Dear Dr Pritchard, 

On behalf of my colleagues and the Academic Editor, Jenny Myers, I am pleased to inform you that we have agreed to publish your manuscript "The Fetal Region-specific Optimised Growth Standard (FROGS) – A fetal and birthweight centile calculator validated in a national population" (PMEDICINE-D-24-02989R3) in PLOS Medicine. I apologize in the delay of providing you with this decision.

However, we do have a few minor requests, that we like you to address:

* Please add this statement to the manuscript's Competing Interests: "RH is a paid statistical consultant on PLOS Medicine's statistical board."

* Statistical reporting: Please revise throughout the manuscript, including tables and figures.

- Please report statistical information as follows to improve clarity for the reader "22% (95% CI [13%,28%]; p</=)".

- Please separate upper and lower bounds with commas instead of hyphens as the latter can be confused with reporting of negative values.

- Please repeat statistical definitions (HR, CI etc.) for each set of parentheses.

Furthermore, before your manuscript can be formally accepted you will need to complete some formatting changes, which you will receive in a follow up email. Please be aware that it may take several days for you to receive this email; during this time no action is required by you. Once you have received these formatting requests, please note that your manuscript will not be scheduled for publication until you have made the required changes.

PRESS

Sincerely, 

Suzanne De Bruijn, PhD 

Associate Editor 

PLOS Medicine